# TOWARDS RELIABLE BACKDOOR ATTACKS ON VISION TRANSFORMERS

## ABSTRACT

Backdoor attacks, which make Convolution Neural Networks (CNNs) exhibit specific behaviors in the presence of a predefined trigger, bring risks to the usage of CNNs. These threats should be also considered on Vision Transformers. However, previous studies found that the existing backdoor attacks are powerful enough in ViTs to bypass common backdoor defenses, *i.e.*, these defenses either fail to reduce the attack success rate or cause a significant accuracy drop. This study investigates the existing backdoor attacks/defenses and finds that this kind of achievement is over-optimistic, caused by inappropriate adaption of defenses from CNNs to ViTs. Existing backdoor attacks can still be easily defended against with proper inheritance from CNNs. Furthermore, we propose a more reliable attack: adding a small perturbation on the trigger is enough to help existing attacks more persistent against various defenses. We hope our contributions, including the finding that existing attacks are still easy to defend with adaptations and the new backdoor attack, will promote more in-depth research into the backdoor robustness of ViTs.

## 1 INTRODUCTION

Vision Transformers (ViTs) (Dosovitskiy et al., 2021; Liu et al., 2021) have demonstrated outstanding performance in various tasks, including image classification (Yuan et al., 2021; Touvron et al., 2022), semantic segmentation (Strudel et al., 2021), and image generation (Hirose et al., 2021; Bao et al., 2022), leading to their widespread popularity. However, strong performance alone is insufficient for ViT to be practically deployable. It must also exhibit security and trustworthiness without posing severe security risks. One of the most notable threats to the security of ViTs is backdoor attacks (Gu et al., 2017; Chen et al., 2017), which implant unex-

Table 1: The performance of FT against Badnets attack for ResNet-18 and ViT-B on CIFAR-10 (Wu et al., 2022).

|      | ResNet18 | ViT-B  |
| ---- | -------- | ------ |
| ASR  | 1.48%    | 8.81%  |
| ACC  | 89.96%   | 42.00% |

pected behaviors inside models, making the victim model produce specific misclassification in the presence of a predefined trigger while maintaining high performance on benign images. While previous studies mainly focus on convolution neural networks (CNNs), there is a growing need for an in-depth investigation of ViTs to help practitioners better understand the potential risks and deploy them more reliably.

After a long arms race between backdoor attack and defense, for CNNs, a relatively simple defense has the potential to make backdoor attacks fail, taking fine-tuning defense and Badnets attack as an example in Table 1, we find that Badnets attack makes the attack success rate (ASR) on ResNet18 only have 1.48% while the benign accuracy (ACC) is 89.96%, which indicates a comprehensive failure of the attack under defense. Contrastingly, ViTs, when subjected to the same attack, display an increased ASR and decreased ACC, implying the success of the attack even under defense. Given that Badnets is model-agnostic, this differential outcome piqued our interest, driving us to explore the underlying disparities between CNNs and ViTs.

Drawing inspiration from Mo et al. (2022), we discerned a crucial observation: 1) CNNs are usually trained by SGD and its fine-tuning defense is also trained by SGD; 2) ViTs are typically trained by AdamW while its fine-tuning defense is trained by SGD (NOT AdamW, inheriting from earliest work (Dosovitskiy et al., 2021), which first introduces optimizers to computer vision). This discrepancy in optimizers raises the possibility that the perceived vulnerability of ViTs (with defense) might be overstated, i.e., the success of attacks on ViTs with defense may be questionable. In this paper, we first conduct a series of experiments to comprehensively investigate the above hypothesis, which is further confirmed that the threat posed to ViTs with defense has been magnified. Upon minor

modifications, ViTs with existing backdoor defense methods demonstrate clear resistance to attacks, mirroring the robustness of CNNs.

To this end, we are wondering whether a more elusive attack exists that can seamlessly sidestep current defenses. Therefore, we analyze backdoored models and further propose a simple yet effective attack. We discover that it is easy for backdoor defenses to detect and utilize the differences in channel activations due to the noticeable difference in the intermediate layers between the inputs with and without triggers. However, we can reduce this difference by adding small perturbations to the triggers during training while keeping triggers unchanged during testing, resulting in more reliable backdoor attacks. Additionally, our method has transferability across different transformer architectures and is effective for both small and large datasets.

In summary, our contributions are summarized as follows:

- We investigate the existing backdoor defenses on ViTs and find the outstanding performance of the backdoor attacks to ViTs is over-estimated due to the inappropriate adaption from CNNs to ViTs. Further, we provide a practical training recipe to improve the defense performance of existing methods and show that existing attacks can not provide reliable performances after defense.

- We propose to add small perturbations to the triggers during training to suppress the difference in the intermediate-level representations between the inputs with and without triggers, resulting in a reliable attack. The proposed method can transfer across various architectures.

- Our contributions, including the finding of existing attacks to current defenses and the development of a new attack, contribute to a reliable baseline for the backdoor robustness of ViTs. We hope it can be a cornerstone of future studies in the backdoor robustness of ViTs.

## 2 RELATED WORK

### 2.1 BACKDOOR ATTACK

Backdoor attacks (Gu et al., 2017; Chen et al., 2017), also known as Trojan attacks, indicate the behaviors of implanting specific malicious behavior into machine learning models, which make the models perform well on benign data while leading to specific misclassifications on inputs containing triggers (*i.e.*, triggered inputs). The adversary usually poisons the training data (Zeng et al., 2021) or controls the training process (Liu et al., 2018b) to achieve this. Typically, a trigger pattern is added to the input image as follows,

$$x_p = (1 - m) \odot x + m \odot t, \tag{1}$$

where $t$ is the trigger pattern and mask $m$ indicates the pixels affected by the trigger pattern. Usually, the adversary re-labels the triggered input as the predefined target class (*i.e.* in a dirty-label setting). Models trained on a mixture of these poisoned data and other benign data are implanted with an unexpected correlation between the trigger pattern and the target class. To improve the stealthiness of the attacks, some studies explored less noticeable trigger designs like the semi-transparent trigger (Chen et al., 2017), the elastic transformed trigger (Nguyen & Tran, 2021), and the input-aware trigger (Nguyen & Tran, 2020). Besides, since incorrect annotation might expose the existence of triggered data, some studies focus on poisoning without re-labeling (clean-label settings) (Turner et al., 2019; Barni et al., 2019; Shafahi et al., 2018). Although most previous backdoor attacks focus on CNNs, researchers have started to focus on backdoor attacks on ViT since their increasing popularity. Although ViTs are reported to be more robust against adversarial attacks (Aldahdooh et al., 2021; Shao et al., 2021) and common corruption (Bai et al., 2021; Bhojanapalli et al., 2021), they are still vulnerable to backdoor attacks (Lv et al., 2021; Subramanya et al., 2022a). Reliable attacks are needed to help practitioners properly understand the risks of backdoor attacks and deploy these models reliably.

### 2.2 BACKDOOR DEFENSE

To mitigate the potential risks caused by backdoor attacks, numerous studies proposed various defense methods, mainly categorized into **defense during training** and **defense after training** based on the stages at which they are applied. Defense during training attempts to mitigate the impact of poisoned data in the training set. Some methods detect and remove poisoned data by treating them as outliers (Chou et al., 2018; Udeshi et al., 2022; Gao et al., 2019), some employ semi-supervised learning to bypass the incorrect correlations (Huang et al., 2022), and others utilize differential privacy to

ensure that a poisoned portion of training data is unable to cause severe results (Miao et al., 2022). Meanwhile, the defense after training directly removes the backdoor behavior inside DNNs. This can be accomplished by fine-tuning the model using a small amount of clean data (Sha et al., 2022) and it can be further enhanced by first pruning the inactivated neuron (Liu et al., 2018a) or encouraging the alignment of attentions (Li et al., 2021) between the student and the teacher network. Since the performances of fine-tuning are easy to suffer a substantial decrease when the data is limited, another popular method is selectively removing neurons related to the backdoor behaviors (Wu & Wang, 2021; Chai & Chen, 2022; Wang et al., 2019): Built upon the observation that the backdoor behavior can be revealed by the adversarial neuron perturbation, ANP (Wu & Wang, 2021) formulates the following min-max problem to expose the malicious neuron:

$$\min_{\mathbf{m} \in [0,1]^n} \left[ \alpha \mathcal{L}_{D_v}(\mathbf{m} \odot \mathbf{w}, \mathbf{b}) + (1-\alpha) \max_{\boldsymbol{\delta}, \boldsymbol{\xi} \in [-\epsilon, \epsilon]^n} \mathcal{L}_{D_v}((\mathbf{m} + \boldsymbol{\delta}) \odot \mathbf{w}, (1 + \boldsymbol{\xi})\mathbf{b}) \right], \quad (2)$$

where $\boldsymbol{\delta}$ and $\boldsymbol{\xi}$ are the perturbations to maximize the cross-entropy loss $\mathcal{L}_{D_v}$ and $\mathbf{m}$ is the mask which adversarially preserves the clean accuracy and covers up the backdoor behavior. Then the neurons corresponding to low mask values are pruned to purify the backdoor model. As an improved approach based on ANP, AWM in (Chai & Chen, 2022) proposes to adopt the element-wise weight masking strategies and perturb the input data instead of the neurons to gain better performances on small networks. This paper primarily focuses on defense after training. Because ViTs demand a large amount of data and extensive training resources, it has become impractical for most practitioners to train ViTs from scratch, making defense after training a more realistic scenario. Previous studies (Wu et al., 2022; Yuan et al., 2023) suggested that directly applying defenses from CNNs to ViTs fails. For example, fine-tuning decreases natural accuracy from 94.58% to 42.00% against the Badnets attack and fine-pruning totally collapses in Yuan et al. (2023). At the meantime, only a few defense methods specially designed for ViT are proposed (Doan et al., 2022; Subramanya et al., 2022b) and their performance is lagging far behind the state-of-the-art defense on CNNs: The adaptive defense proposed in (Zheng et al., 2022) only decreases the ASR of TrojViT (a ViT-specific attack) to 77.13% and the patch processing method in (Doan et al., 2022) fails to detect 33.2% backdoor examples on CIFAR-10. It seems that existing attacks can already obtain outstanding performances on resisting defense for ViTs. However, in this paper, after re-investigating various backdoor defenses with ViTs, we reveal that the achievement obtained by previous attacks is not reliable. Furthermore, we provide a reliable attack, based on the empirical observation of the channel activations of ViTs. It might help future research on backdoor robustness with ViTs.

## 3    The Vulnerability of ViTs (with defense) to Existing Attacks

In this section, we reevaluate the perceived susceptibility of ViTs to prevailing backdoor attacks when equipped with potential defenses. We primarily consider two categories of defenses: one is fine-tuning-based, including Fine-Tuning (FT) (Sha et al., 2022), Fine-Pruning (FP) (Liu et al., 2018a), and Neural Attention Distillation (NAD) (Li et al., 2021), and the other is pruning-based, including Adversarial Neuron Pruning (ANP) (Wu & Wang, 2021) and Adversarial Weight Masking (AWM) (Chai & Chen, 2022).

### 3.1    Basic Settings

Here, we train a backdoored ViT-B (Dosovitskiy et al., 2021) with various attack methods. Specifically, we initialize the model with a pre-trained weight (Wightman, 2019) on the ImageNet-1k (Deng et al., 2009) and then fine-tune it on CIFAR-10[1] (Krizhevsky et al., 2009). Note that a portion of CIFAR-10 training data is contaminated to implant the backdoor behavior, *i.e.*, some images are added with the trigger pattern and are re-labeled as the target class if expected. We apply four commonly-used attack methods: 1) Badnets (Gu et al., 2019), 2) Blend (Chen et al., 2017), 3) CLB (Turner et al., 2019), and 4) SIG (Barni et al., 2019). Their trigger design and poisoning method in the original paper are kept. To accommodate the input size of ViT, we first add triggers to CIFAR-10 images ($32 \times 32$) and then resize them to a larger size ($224 \times 224$). For detailed information, please refer to Appendix A. Here, we use accuracy (ACC) to indicate the classification performance on benign data, and attack success rate (ASR), the percentage of triggered input being classified as the target class, to indicate the attack performance. Note that we will remove the inputs whose ground-truth label is the target class, and thus, a successful defense should make ASR as low as 0.

---

[1]Ony 95% of the original training data on CIFAR-10 are used to train the backdoored model, and the remaining data are kept for defense.

Table 2: The comparison between SGD and AdamW optimizer on FT. Here, AvgDrop represents the average drop of four attacks on ASR/ACC after performing FT.

| Attack | ACC | | | ASR | | |
|---|---|---|---|---|---|---|
| | No defense | SGD | AdamW | No defense | SGD | AdamW |
| Badnets | 97.85 | 58.74 | 93.79 | 100.00 | 3.40 | 2.51 |
| Blend | 97.85 | 94.33 | 93.30 | 100.00 | 13.49 | 4.91 |
| CLB | 97.83 | 94.60 | 94.06 | 96.23 | 10.49 | 1.33 |
| SIG | 97.50 | 51.56 | 93.51 | 90.57 | 2.23 | 1.40 |
| AvgDrop | - | 22.95 | 4.10↓ | - | 89.30 | 94.16↑ |

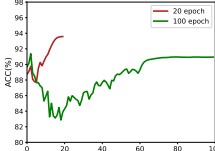 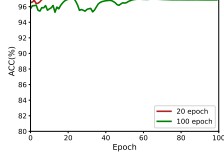 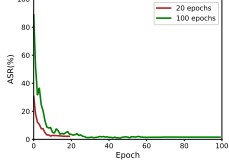 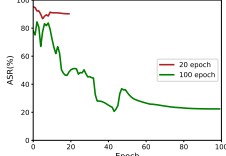

(a) The curve of ACC for various epochs. (*Left*: ViT-B, *Right*: ConvNeXt-B)

(b) The curve of ASR for various epochs. (*Left*: ViT-B, *Right*: ConvNeXt-B)

Figure 1: The Effects of various epochs on ViT-B and ConvNeXt-B for FT.

## 3.2 ViTs with Fine-tuning-based Defense

Fine-tuning is one of the most basic and model-agnostic defenses. However, as discussed in Section 1, directly inheriting fine-tuning-based defense strategies from CNNs can potentially lead to suboptimal outcomes. Here, we investigate several factors, including optimizers and training epochs, which may impact defense performance.

**Optimizers.** SGD is the commonly used optimizer for both training and fine-tuning for CNNs, while for ViTs, the first work (Dosovitskiy et al., 2021) introducing Transformers to computer vision, adopts AdamW for pre-training and SGD for fine-tuning. Notably, prior work (Wu et al., 2022) on backdoor defense naturally inherit this strategy and observes notably diminished accuracy across multiple backdoor attacks. This discrepancy in optimizers motivates us to study the potential influence of optimizers on backdoor defense. The initial learning rates for SGD and AdamW are set to 0.02 and 3e-4, respectively. For the other parameters in AdamW, we use the common settings of the original ViTs (refer to Appendix B for details). Table 2 illustrates the experimental fine-tuning (FT) results against various backdoor attacks. For the results on FP and NAD, please refer to Appendix C. We find that SGD exhibited significant instability on ViTs. Even for the same model, when defending against Blend and CLB, it achieves more than 90% of ACC. However, for BadNet and SIG, ACC decreases to less than 60%. In contrast, AdamW consistently achieves high ACC and low ASR using the same hyper-parameter configuration. Therefore, simply using SGD for backdoor defense on ViTs will yield highly unstable performance. We recommend employing AdamW for defense purposes.

**Fine-tuning Epochs.** Typically, ViTs require more epochs to train from scratch, which leads us to explore whether the number of epochs would have different effects on the defense with CNNs and ViTs. Here, we fine-tune a backdoored ViT-B for either 20 or 100 epochs. As a comparison, we simultaneously fine-tune a backdoored ConvNeXt-B (Liu et al., 2022), which has a similar number of parameters. In Figure 1, we find that, for ConvNeXt-B, fine-tuning for more epochs reduces ASR notably while slightly decreasing ACC. However, for ViT-B, more epochs cause a significant ACC drop, making the model unusable. Due to ViT's sensitivity to the number of training epochs when we only have a limited amount of clean data for defense, we recommend using fewer epochs. For the final ACC and ASR for all fine-tuning-based defenses, please refer to Appendix D.

## 3.3 ViTs with Pruning-based Defense

Pruning is also a typical defense approach, which attempts to remove backdoor-related neurons/channels and is severely impacted by the architectures. In previous studies, pruning-based methods have achieved excellent robustness against backdoor attacks with CNNs (Wu & Wang, 2021; Chai & Chen, 2022). However, when we directly apply these methods to ViTs, we find that they are unable to effectively defend as shown in Table 3. Specifically, ANP fails to reduce ASR and

Table 3: The Performance of ANP and AWM with or without ViTs adaptation.

| Metric | Setting | Before | ANP | ANP (ViTs adapted) | AWM | AWM (ViTs adapted) |
|--------|---------|--------|------|--------------------|------|--------------------|
| ACC | Badnets | 97.85 | 97.85 | 94.26 | 85.98 | 95.02 |
|     | Blend   | 97.85 | 97.85 | 92.70 | 83.29 | 95.08 |
|     | CLB     | 97.83 | 97.83 | 95.71 | 85.67 | 95.60 |
|     | SIG     | 97.50 | 97.50 | 92.60 | 87.22 | 94.58 |
| ASR | Badnets | 100.00 | 100.00 | 1.34 | 1.24 | 0.71 |
|     | Blend   | 100.00 | 100.00 | 23.7 | 2.03 | 1.70 |
|     | CLB     | 96.23 | 96.23 | 12.71 | 3.48 | 1.52 |
|     | SIG     | 90.57 | 90.57 | 1.48 | 1.16 | 3.87 |

cannot remove the backdoor-related neurons. Besides, although AWM reduces ASR, it also severely decreases ACC, making the model unusable. To explore the potential reason, we look deeply at the implementation of ANP and AWM and find that ANP actually prunes channels inside norm layers rather than neurons inside convolutional layers. This is because, in CNNs, each neuron is typically surrounded by at least one norm layer[2]. However, in ViT, many norm layers are removed, and norm-layer-based pruning only influences part of neurons and limits the defense performance. Meanwhile, AWM utilizes element-wise masks for optimization, whose number of parameters is the same as the total number of parameters of ViT. Since ViTs are typically larger, AWM encounters the severe overfitting issue, leading to low accuracy. Therefore, to make pruning methods applicable to ViTs, selecting appropriate granularity and pruning locations is necessary. Here, we recommend directly pruning all channels of linear projection inside both attention and MLP layers, which provides better coverage than ANP and requires fewer parameters compared to AWM. This modification decreases ASR notably and keeps ACC high.

## 4 PROPOSED BACKDOOR ATTACKS

Following the above analysis, existing defense methods (ViTs adapted) successfully defend against backdoor attacks in ViTs, just as they do in CNNs. Here, we want to explore whether there exist new backdoor attacks to beat the newly adapted defense on ViTs.

To obtain a better insight into why defense methods can detect and remove backdoor behaviors, we investigate the per-channel activations before the MLP layers in ViT. We illustrate the average activations of all channels for a backdoored ViT-B on triggered and benign inputs from the CIFAR-10 test set, respectively. For clarity, we reorganize the channels based on their average activations, arranging them from largest to smallest with respect to average activations on benign data. In Figure 2, we find a significant activation difference between benign and triggered inputs, which is easy to capture. Further, we compare the average activation of all channels for models purified by FT and AWM, and find that benign and triggered inputs have similar average activation after defense. This suggests that the naive trigger design (usually predefined universal patterns) for current backdoor attacks results in a significant difference between benign and triggered data, revealing attack information to possible defenders. Next, we will study whether we could improve the trigger design to escape defenses. The general process of our attack is summarized in Figure 3 and we term it as the Channel Activation attack in ViT (CAT).

**Adversarial Loss.** Based on our observation, a good trigger design is expected to avoid noticeable channel activation differences between benign and triggered inputs. Therefore, we require additional backdoor discriminators (BD) to clarify whether the training input has the predefined trigger during the training. Specifically, we denote the feature extractor of the backdoored model as $g(\cdot)$[3], and the backdoor discriminator $d_i(g(x))$ uses the intermediate feature of the $i$-th layer to discriminate whether the input $x$ has the trigger pattern. During backdoor training, we also train these backdoor discriminators of the last $n$ layers, *i.e.*, $d_i(g(x)), i = L - n + 1, \cdots, L$. After training, we could use these backdoor discriminators to generate adversarial perturbations on the trigger pattern to minimize the activation difference between benign and triggered inputs. Meanwhile, naive difference minimization might make the model classify triggered inputs as a non-target label, leading to the failure of backdoor attacks. To address this issue, we introduce additional target classifiers

---

[2]Specifically, for Preact-ResNet, the norm layer is always located before the neuron; for ResNet, it is located after the neuron

[3]In our method, the extractor will return intermediate features from all layers.

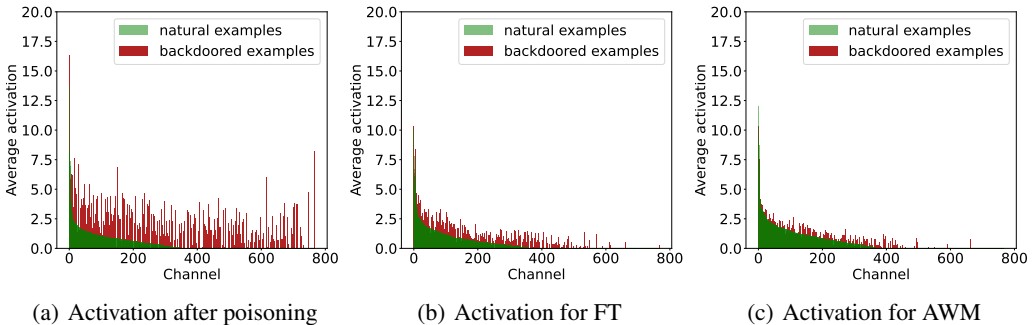

Figure 2: The average activations for different channels before (a) and after the backdoor defense (b)-(c). The activations are sorted in descending order of the activations on natural samples.

$f_i(\boldsymbol{g}(\boldsymbol{x}))$ (TC), which uses the intermediate feature of the $i$-th layer to make classification between benign samples, *i.e.*, classifying the benign input as the ground-truth label. Similar to the backdoor discriminator, we also train these clean classifiers of the last $n$ layers, *i.e.*, $f_i(\boldsymbol{g}(\boldsymbol{x})), i = L - n + 1, \cdots, L$ during training. In conclusion, we craft adversarial perturbation via maximizing the following loss,

$$\mathcal{L}(\boldsymbol{\delta}) = \sum_{i=L-n+1}^{L} (1 - \gamma) \cdot \ell\big(d_i(\boldsymbol{g}(\boldsymbol{x} + \boldsymbol{m} \odot \boldsymbol{\delta})), y_{\text{bd}}\big) - \gamma \cdot \ell\big(f_i(\boldsymbol{g}(\boldsymbol{x} + \boldsymbol{m} \odot \boldsymbol{\delta})), y_{\text{tc}}\big), \quad (3)$$

where $y_{bd}$ is the label for the backdoor discriminator, *i.e.*, 1 for triggered data and 0 for benign data. $y_{tc}$ is the label for the target classifier as the adversary expects, *i.e.*, the ground-truth label for benign input, and the target label for triggered input. Here $\gamma$ is a trade-off coefficient to balance the effect between TC and BD.

**Generation Steps.** Since the nonlinearity of ViTs, it is mathematically infeasible to obtain the exact solution for Equation 3. However, we can use the projected gradient descent (PGD) (Madry et al., 2018) from the normal adversarial attacks to craft the perturbations on the trigger pattern as follows:

$$\boldsymbol{\delta} \leftarrow \boldsymbol{m} \odot \Pi_\epsilon\big(\boldsymbol{\delta} + \alpha \cdot \text{sign}(\nabla_{\boldsymbol{\delta}}\mathcal{L}(\boldsymbol{\delta}))\big), \quad (4)$$

where $\boldsymbol{m}$ is the mask for triggers, $\odot$ is the Hadamard product, and $\Pi_\epsilon(\cdot)$ is the projection function,

$$\Pi_\epsilon(\boldsymbol{\delta}) = \frac{\epsilon}{\|\boldsymbol{\delta}\|_2}\boldsymbol{\delta}. \quad (5)$$

**Random Masking of Perturbation.** In practical situations, the adversary has no access to model architecture and its parameters. Usually, the adversary expects to craft these perturbations from models with known parameters and structure (**source model**) to attack these unknown models (**target model**). The generated perturbations in this situation are expected to be effective across various architectures. Unfortunately, different ViTs could have various patch sizes for splitting, leading to differences in the scale of sensitive features. This might cause low transferability across architectures. Therefore, we propose a method termed Random Masking of Perturbation (RMP). In each step during crafting adversarial perturbations, we first split perturbation with $k$ patches and randomly drop a predefined percentage of perturbation patches. This can create features of varying scales manually and make the perturbations effective for kinds of ViTs with different patch-splitting approaches.

## 5 EXPERIMENTS

### 5.1 MAIN RESULTS

**Settings:** We evaluate the performances of our methods in two scenarios. 1) **White-box**: the target model and source models have the same architectures and backdoor training from the same pre-trained model. 2) **Black-box**: the architectures of the target model and the source model are different. We choose ViT-B as the source model and five ViT variants, including ViT-B, DeiT-S (Touvron et al., 2021a), Swin-B (Liu et al., 2021), Cait-S (Touvron et al., 2021b) and XciT-S (Ali et al., 2021) as our target models. In our experiments, we choose the last two layers (*i.e.*, $n = 2$) to add BD and TC modules. For the perturbation generation step, the adversarial attack is $l_2$ bounded PGD-10 with

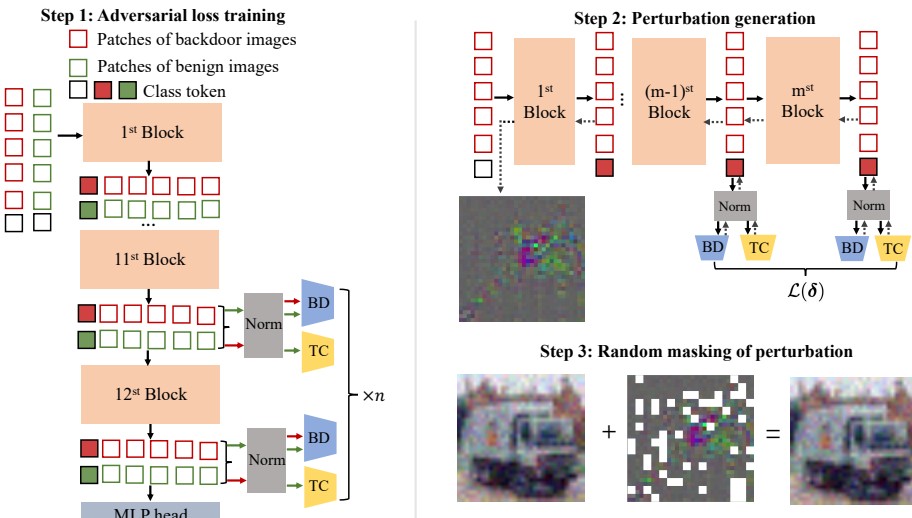

Figure 3: The illustration of our proposed attack. We illustrate our attack by taking ViT-B as an example. *left:* Using the existing poisoned dataset, we also train the BD and TC simultaneously during backdoor training (**Step 1**). *right:* When the training is over, we perform adversarial attacks on the BD and TC modules to generate adversarial perturbation (**Step 2**). In each step during crafting adversarial perturbations, we manually mask some patches of perturbation to better poisoned ViTs (**Step 3**).

budget $16/255$, step size $4/255$, and the trade-off parameter $\gamma$ is set to 0.6. For random masking of perturbation, we split the perturbation into multiple small pieces, each of which has the shape of $2 \times 2$. The percentage of dropped patches is set to 0.1 and 0.05 for the whole-image patch and trigger-based path, respectively. For other hyperparameters, we keep in line with Section 3. All experiments are performed on CIFAR-10. The performances of our methods CAT on ASR are summarized in Table 4. For ACC, please refer to Appendix E.

**Results:** First, when no defenses are performed, CAT will obtain a comparable ASR compared to the vanilla settings. In most cases, it even can gain better performance. For example, our method increases the ASR of SIG attacks from 90.57% to 91.19% on ViT-B. Second, for the post-defense situation, CAT can achieve higher ASR in a novel margin. For example, under the white-box setting, it increases the ASR from 2.51% to 66.72% against the badnets attack for FT. In the black-box settings, the ASR of SIG attacks increases from 3.30% to 13.81% on DeiT-S for the AWM defenses. As for ACC, the results in Appendix E show that CAT will obtain comparable ACC compared to the vanilla attack. It indicates that our method will only enhance the ASR without compromising the performance of the benign image classification. When comparing the results across architectures, we notice that almost all defenses obtain worse performances on Cait-S and XciT-S. We conjecture the reason may be that both architectures adopt the multi-head class attention layers as their components which will more efficiently extract backdoor information from the input data. It increases the difficulty of performing defense and our attacks can further improve the attack performances.

## 5.2 Performance on ImageNet with Comparisons with ViT-specific Methods

Attribute to the highly flexible multi-head self-attention mechanism, ViTs can outperform CNNs when millions of data are provided. Thus in this section, we not only evaluate the performance of our attack on ImageNet (Deng et al., 2009) but also compare it with existing ViT-specific attacks to illustrate its superiority. Here we only report the results after combining badnets and blend attacks because the clean-label attacks will fail for only at-most poisoning 0.1% of training data. More details of our experimental setup are summarized in Appendix F. Because of the huge computational costs, both the source model and the target model are selected as ViT-B. In addition to the model-agnostic attacks mentioned in the previous sections, we also include two **ViT-specific attacks**: the Trojan Insertion attack in ViT (TrojViT) (Zheng et al., 2022) and the Data-free Backdoor Injection Attack (DBIA) (Lv et al., 2021) as baselines. In addition to studying ViT-specific attacks, we also want to investigate whether our methods can better evade the ViT-specific defense, such as Attention Blocking (AB) (Subramanya et al., 2022b). AB identifies the triggers through the attention roll-out (Abnar &

Table 4: ASR (%) of our proposed attack with different ViT variants on the CIFAR-10 dataset. The best results are in **bold**.

| Defense | Attack | Vanilla | | | | | CAT | | | | |
|---|---|---|---|---|---|---|---|---|---|---|---|
| | | ViT-B | DeiT-S | Swin-B | Cait-S | XciT-S | ViT-B | DeiT-S | Swin-B | Cait-S | XciT-S |
| No defense | BadNets | 100.00 | 100.00 | 100.00 | 100.00 | 100.00 | 100.00 | 100.00 | 100.00 | 100.00 | 100.00 |
| | Blend | 100.00 | 100.00 | 100.00 | 100.00 | 100.00 | 100.00 | 100.00 | 100.00 | 100.00 | 100.00 |
| | CLB | 96.23 | 95.28 | 84.86 | 85.71 | 100.00 | 94.57 | 94.04 | 90.23 | 92.21 | 100.00 |
| | SIG | 90.57 | 84.77 | 94.99 | 80.93 | 94.21 | 91.19 | 88.28 | 97.77 | 82.26 | 96.17 |
| FT | BadNets | 2.51 | 86.57 | 33.27 | 85.21 | 81.89 | 66.72 | 96.30 | 57.53 | 99.79 | 90.11 |
| | Blend | 4.91 | 13.06 | 73.32 | 93.60 | 86.34 | 38.53 | 49.14 | 95.96 | 99.79 | 98.57 |
| | CLB | 1.33 | 33.61 | 7.89 | 34.73 | 89.74 | 12.32 | 70.53 | 10.11 | 53.73 | 93.73 |
| | SIG | 1.40 | 25.83 | 46.90 | 20.47 | 58.91 | 10.99 | 42.14 | 50.57 | 36.77 | 77.34 |
| FP | BadNets | 0.91 | 33.98 | 11.49 | 15.80 | 6.37 | 27.90 | 45.09 | 19.52 | 19.96 | 14.39 |
| | Blend | 0.73 | 3.82 | 2.48 | 43.09 | 23.82 | 12.49 | 14.73 | 22.67 | 90.27 | 29.50 |
| | CLB | 1.70 | 3.87 | 2.54 | 1.59 | 13.99 | 26.88 | 16.52 | 5.56 | 6.12 | 20.49 |
| | SIG | 0.81 | 2.26 | 3.81 | 8.96 | 13.22 | 9.68 | 14.79 | 5.49 | 19.23 | 16.43 |
| NAD | BadNets | 1.57 | 84.21 | 47.56 | 95.09 | 86.08 | 86.50 | 97.88 | 73.37 | 97.87 | 98.48 |
| | Blend | 8.94 | 57.01 | 85.31 | 97.68 | 91.78 | 61.93 | 64.64 | 98.90 | 98.27 | 93.12 |
| | CLB | 7.27 | 52.12 | 13.83 | 95.09 | 86.08 | 13.30 | 65.93 | 16.48 | 97.87 | 98.48 |
| | SIG | 3.60 | 25.81 | 41.10 | 19.01 | 11.30 | 9.07 | 36.87 | 53.47 | 24.62 | 58.21 |
| ANP | BadNets | 1.34 | 92.62 | 47.62 | 91.6 | 75.48 | 51.09 | 93.97 | 48.80 | 98.81 | 96.40 |
| | Blend | 23.70 | 92.82 | 97.04 | 99.97 | 98.77 | 92.23 | 96.71 | 100.00 | 100.00 | 99.01 |
| | CLB | 12.71 | 75.76 | 4.10 | 1.01 | 88.10 | 14.01 | 81.01 | 1.81 | 19.40 | 90.51 |
| | SIG | 1.48 | 67.61 | 79.61 | 64.17 | 10.89 | 67.57 | 72.51 | 83.18 | 71.24 | 43.39 |
| AWM | BadNets | 0.71 | 2.71 | 4.79 | 0.90 | 2.31 | 6.78 | 6.64 | 12.76 | 10.57 | 16.11 |
| | Blend | 1.70 | 1.27 | 0.32 | 36.00 | 88.43 | 26.22 | 5.12 | 27.62 | 57.72 | 94.56 |
| | CLB | 1.52 | 2.19 | 3.16 | 0.91 | 26.84 | 4.40 | 5.42 | 6.74 | 2.66 | 40.71 |
| | SIG | 3.87 | 3.30 | 29.83 | 16.79 | 35.99 | 38.59 | 13.81 | 59.82 | 23.22 | 96.05 |

Table 5: ASR (%) of our attack on ImageNet dataset. The higher ASR is in **bold**.

| Attack | Before | FT | FP | NAD | ANP | AWM | AB |
|---|---|---|---|---|---|---|---|
| TrojViT | 91.08 | 0.14 | 0.11 | 0.16 | 0.46 | 0.18 | - |
| DBIA | 99.58 | 0.09 | 0.07 | 0.10 | 0.10 | 0.05 | - |
| Badnets | 100.00 | 27.75 | 3.67 | 26.82 | 18.30 | 24.32 | 3.84 |
| Badnets+CAT | 100.00 | **51.35** | **14.17** | **28.75** | **44.36** | **81.98** | **12.76** |
| Blend | 100.00 | 18.44 | 1.01 | 6.71 | 19.79 | 39.63 | 100.00 |
| Blend+CAT | 100.00 | **27.83** | **3.17** | **13.44** | **48.49** | **71.29** | **100.00** |

Zuidema, 2020) and masks them with a $30 \times 30$ patches. The hyperparameter settings of ViT-specific attacks or defenses are exactly the same as those in the original paper (Please refer to Appendix F for details). The ASR and ACC of our experiments on ImageNet are summarized in Table 5 and Appendix G, respectively.

First, similar to the results on CIFAR-10, the results reveal that CAT can help existing attacks better bypass the adapted defenses. For example, our approach boosts the ASR of Badnets from 24.32% to 81.98% after applying AWM. In addition, compared to the existing ViT-specific backdoor attacks, our method also shows its superior performance: Both TrojViT and DBIA only obtain less than $< 1\%$ ASR after performing fine-tuning-based or pruning-based defense which is quite lower than those for our attack. In addition, for ViT-specific defense, our method also obtains better performance: the gains on ASR are observed after combining Badnets with CAT. We conjecture this is because our attack reduces the anomalous behavior of backdoor samples on ViTs by introducing benign features. This increases the difficulty of detecting them from the poison dataset. AB totally fails to defend Blend or CAT+Blend because it only masks a patch of images which will be less effective when encountering the whole-image attack, *i.e.* Blend.

## 5.3 ABLATION STUDY

For our proposed method CAT, there are two key components: one is to perform adversarial attacks on triggers (PA), and the other is to randomly mask patches of perturbation (RMP). To evaluate the contribution of each component, we test the performances under three combinations: 1) the vanilla backdoor attacks, 2) backdoor attacks with PA, 3) backdoor attacks with both PA and RMP. Considering both white-box and black-box settings, we select ViT-B and Swin-B as the target models. We select FP and AWM to evaluate the performances of backdoor attacks since they show the most promising performances in Table 4. Other configurations are the same as those in section 5.1. We summarize the ASR for all combinations in Table 6. It reveals that PA can improve the ASR for both

Table 6: The ASR for different combinations of our technique. The better result is in **bold**.

| | Attack | ViT-B | | | | Swin-B | | | |
|---|---|---|---|---|---|---|---|---|---|
| | | Badnets | Blend | CLB | SIG | Badnets | Blend | CLB | SIG |
| FP | Vanilla | 0.91 | 0.73 | 1.70 | 0.81 | 11.49 | 2.48 | 2.54 | 3.81 |
| | +PA | 14.54 | 6.52 | 7.50 | 7.04 | 15.19 | 14.97 | 3.40 | 3.82 |
| | +PA+RMP | **27.90** | **12.49** | **26.88** | **9.68** | **19.52** | **22.67** | **5.56** | **5.49** |
| AWM | Vanilla | 0.71 | 1.70 | 1.52 | 3.87 | 4.97 | 0.32 | 3.16 | 29.83 |
| | +PA | 4.78 | 23.26 | 2.48 | 21.52 | 11.32 | 26.32 | 4.39 | 47.87 |
| | +PA+RMP | **6.78** | **26.22** | **4.40** | **38.59** | **12.76** | **27.62** | **6.74** | **59.82** |

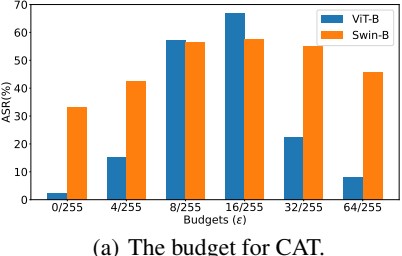

(a) The budget for CAT.

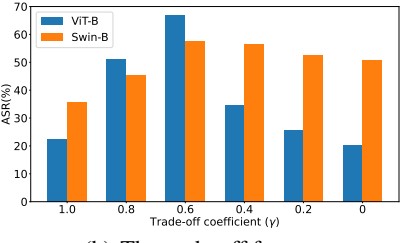

(b) The trade-off factor $\gamma$.

Figure 4: The effect of hyperparameters to the performances of our method.

ViT-B and Swin-B. Applying PA and RDP together can gain higher ASR, *e.g.*, for the white-box setting, the gain of PA for FP on badnets attack is 13.63%, performing PA and RMP both can further improve the ASR for 26.99%. Similar results are also observed for the black-box settings.

### 5.4 HYPERPARAMETER ANALYSIS

In this section, we test the effect of hyperparameters on our proposed methods. Taking Badnets attacks as an example, we report the ASR after performing fine-tuning (FT) for ViT-B and Swin-B.

**Attack budget**: Recalling that in Section 4, we craft the adversarial samples to reduce the differences in features between the backdoor and benign data. The previous works reveal that the strength of the attacks plays a vital significance in the adversarial region. Therefore, we first investigate the effect of the attack strength $\epsilon$ on the performance of our method. As shown in Figure 4 (a), the ASR of our method increases when we increase the budget. This is because more and more features on the triggers that mismatches the benign data are removed. However, when the attack is too strong ($\epsilon > 16/255$), the performance of our method will decrease because it makes it too hard for the network to learn backdoor information from the data.

**Trade-off coefficient**: $\gamma$ is another important hyperparameter for our method. As shown in Figure 4 (b), the results illustrate that the adversarial information from both additional modules: the backdoor discriminator and the target classifier can improve the ASR ($\gamma = 0$ or 1.0). However, mixing the information from both modules can gain better performance. When $\gamma = 0.6$, our method achieves the best performance by simultaneously enhancing the information of the target class while eliminating the irrelevant features on the triggers.

### 6 CONCLUSION

In this paper, we conduct a comprehensive evaluation of backdoor methods on ViTs and show that the illustration of success achieved by current attacks to ViTs is due to inappropriate adaption of defense from CNNs to ViTs. We further provide some training recipes to correctly evaluate the attack, including using AdamW rather than SGD, using fewer epochs, and selecting appropriate granularity for pruning. Our results demonstrate that existing attacks can not provide reliable performance after defense. Therefore, we investigate why the defense method easily removes backdoor behavior and find a huge difference in channel activation in intermediate layers with commonly used predefined triggers. Inspired by this, we propose a more reliable attack by adding special adversarial perturbations into the trigger pattern to avoid noticeable channel activation differences between benign and triggered input. We hope our method, including the proposed recipes in ViTs and the new attack method, could be a cornerstone of future studies on the backdoor robustness of ViTs.

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

## A  DETAILED SETTINGS FOR BACKDOOR ATTACK

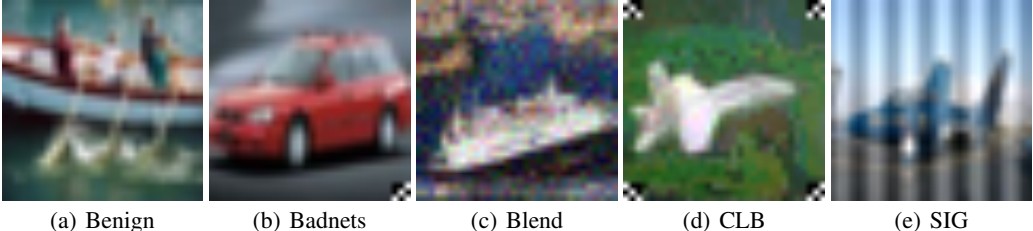

| (a) Benign | (b) Badnets | (c) Blend | (d) CLB | (e) SIG |

Figure 5: Examples for the benign and backdoor images in the poisoned training set.

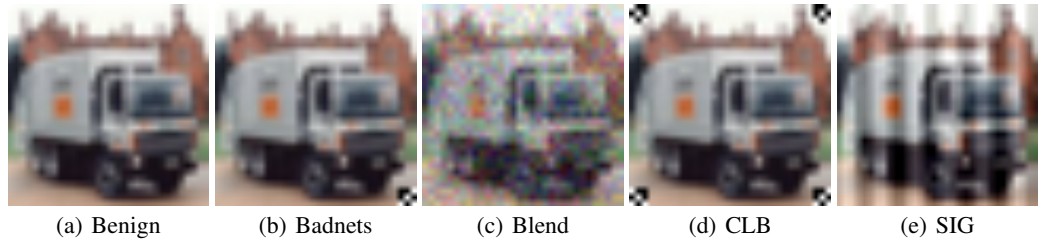

| (a) Benign | (b) Badnets | (c) Blend | (d) CLB | (e) SIG |

Figure 6: Examples for the benign and backdoor images in the poisoned test set.

This section provides detailed information about the settings for the backdoor attacks. As demonstrated in Section 3.1, we first pre-train the ViT-B on ImageNet-1k and finetune the network on the poisoned dataset using AdamW optimizer for 20 epochs with a learning rate of $0.0001$. Simple data augmentations, including random crop with padding and horizontal flipping, are adopted for backdoor training. We assign the Class 0 ("airplane") of the CIFAR-10 dataset as the target class for backdoor attacks. Examples of benign and backdoor images in the training set and poisoned test set are shown in Figure 5 and Figure 6. All experiments are performed on the NVIDIA 3090 GPUs. The implementation details of each attack are summarized as follows:

**Badnets:** Following the original paper (Gu et al., 2019), we take a $3\times3$ checkerboard as the trigger. As shown in Figure 5(b), the trigger is placed at the bottom right corner of the original image. Given the target class, 5% of images from the other classes are attached with the trigger and re-labeled as the target class. For ViT-B, we obtain the ACC of 97.85% and ASR of 100.00%.

**Blend:** For Blend attack, we take the Gaussian noise ($t$) as the trigger. In particular, the trigger has the same size as the original image. For the benign image $x$, the poisoned image can be given as $x_p = (1 - \alpha) \cdot x + \alpha \cdot t$. In contrast to the definition shown in Section 2.1, $\alpha \in [0, 1]$ denotes as the blending rate between the benign image and the trigger pattern. Following the original paper (Chen et al., 2017), $\alpha$ is set to 0.2. Examples of poisoned images in the training and test set are shown in Figure 5(c) and Figure 6(c). Same as Badnets attack, 5% images from the other classes are attached with the trigger pattern and relabeled as Class 0. For ViT-B, we achieve the ACC of 97.85% and ASR of 100.00%.

**CLB:** We select 80% benign images from the target class for data poisoning. Next, we perform a 100-step PGD attack on the selected images using a pre-trained robust model [4]. For the hyperparameter settings, we follow the original paper with the budget $16/255$ and the step size of $2.4/255$. As shown in Figure 5(d), we attach the trigger, a four-corner $3 \times 3$ checkerboard, on these selected images. The poisoned training set combines these poisoned images and the remaining benign images from all classes. For ViT-B, we obtain the ACC of 97.83% and ASR of 96.23%.

**SIG:** We follow the original work in (Barni et al., 2019), which adopts the sinusoidal signal as the trigger. We also select 80% benign images from the target class for data poisoning. The strength $\Delta$ and frequency $f$ for SIG attack are set to 40 and 6 respectively following previous studies (Wu et al., 2022; Barni et al., 2019). Examples of the poisoned images are shown in Figure 5(e) and Figure 6(e). For ViT-B, we obtain the ACC of 97.50% and ASR of 90.57%.

---

[4] https://github.com/yaircarmon/semisup-adv

Table 7: The effect of optimizer on FP and NAD. AdamW gains higher ACC and lower ASR than SGD.

| (a) ACC | | | | | | (b) ASR | | | | | |
|---|---|---|---|---|---|---|---|---|---|---|---|
| Attack | | SGD | | AdamW | | Attack | | SGD | | AdamW | |
| | No defense | FP | NAD | FP | NAD | | No defense | FP | NAD | FP | NAD |
| Badnets | 97.85 | 93.17 | 57.59 | 93.52 | 93.77 | Badnets | 100.00 | 0.90 | 4.24 | 0.91 | 1.57 |
| Blend | 97.85 | 93.41 | 94.27 | 92.59 | 94.09 | Blend | 100.00 | 9.67 | 48.57 | 0.73 | 8.94 |
| CLB | 97.83 | 27.20 | 94.31 | 93.22 | 93.88 | CLB | 96.23 | 8.21 | 10.15 | 1.70 | 7.27 |
| SIG | 97.50 | 77.34 | 94.31 | 93.88 | 93.86 | SIG | 90.57 | 1.93 | 5.00 | 0.81 | 3.60 |
| AvgDrop | - | 24.98 | 12.91 | 4.46↓ | 3.86↓ | AvgDrop | - | 91.53 | 79.71 | 95.66↑ | 91.36↑ |

## B    DETAILED SETTINGS FOR BACKDOOR DEFENSE

This section provides detailed information on the backdoor defenses applied in this paper. The settings of each defense are summarized as follows:

**FT:** We use AdamW (Loshchilov & Hutter, 2018) optimizer, the most popular optimizer for ViTs, to fine-tune the backdoor ViTs for 20 epochs with a learning rate of 3e-4 and a weight decay of 0.2. In addition, we adopt the cosine learning rate schedule. Same as backdoor training, only simple data augmentations, including random crop with padding and horizontal flipping, are used to retain the clean accuracy better and avoid the increasing ASR of whole-image backdoor attacks caused by strong data augmentation as discussed in section 3.

**FP:** FP (Liu et al., 2018a) first prunes the last layer of CNNs by a predefined pruning threshold and then fine-tune the network on the clean subset of data. Similarly, we prune the last linear projection layer of transformer encoder blocks in ViTs. For the pruning partition threshold, we use *the tolerance of clean accuracy reduction* to limit the maximum drop of the benign accuracy following (Wu et al., 2022). In this paper, we set it to 0.9. The other settings are the same as the original paper (Liu et al., 2018a).

**NAD:** NAD (Li et al., 2021) first makes two copies of the original backdoor models, referred to as the teacher model and student model respectively. Next, NAD fine-tunes the teacher model with the vanilla FT. Finally, the finetuning of the student model is guided through neural attention transfer from the teacher model. For the hyperparameter setting, we mainly keep in line with (Wu et al., 2022) except for two differences: we train the student network for 20 epochs using the AdamW optimizer instead of hundreds of epochs with SGD optimizer. The above changes are made because of the observation shown in Appendix C and Appendix D.

**ANP:** Wu et al. (Wu et al., 2020) observe that backdoor models are prone to output the target labels when the neurons are perturbed by the adversarial perturbations. Inspired by this, they propose to optimize the mask of each neuron, a continuous value in $[0, 1]$, under adversarial neuron perturbations and then prune neurons whose mask values are lower than the threshold, *i.e.*, hardening the continuous mask values as binary masks. In this paper, we use the same settings as the original paper except for applying 4000 iterations to avoid under-convergence of large models like ViTs (longer than the 2000 iterations for CNNs in the original paper). Compared to the hardened masks (pruned) applied in their original paper, we find that soft masks, continuous mask values without hardening, can preserve ACC better and decrease ASR further. Thus, we apply soft masks in this paper, and these masks are applied to the channels of linear projection.

**AWM:** Compared to ANP, AWM (Chai & Chen, 2022) makes two improvements on CNNs. The authors apply soft element-wise weight masking instead of neuron pruning (hardened mask values) to avoid over-cutting beneficial information. Besides, they perturb the data instead of the neurons to utilize the training data more efficiently. When applied to ViTs, we mask the channel of the linear projection, similar to ANP. The other hyperparameters are the same as the original paper (Chai & Chen, 2022) without turning.

## C    THE EFFECT OF OPTIMIZER ON FP AND NAD

In this section, we compare the performance of SGD and AdamW on the other two fine-tuning-based methods, FP and NAD, following the settings in section 3.2. As shown in Table 7, the results demonstrate that, compared to SGD, AdamW always performs better on FP and NAD. For example,

Table 9: ACC (%) of our attacks with different ViT variants on the benchmark dataset. The best results are in **bold**.

| Defense | Attack | Vanilla | | | | | Ours | | | | |
|---|---|---|---|---|---|---|---|---|---|---|---|
| | | ViT-B | DeiT-S | Swin-B | Cait-S | XciT-S | ViT-B | DeiT-S | Swin-B | Cait-S | XciT-S |
| No defense | BadNets | 97.85 | 97.67 | 98.53 | 98.47 | 97.83 | **98.18** | **97.75** | **98.69** | 98.35 | **97.90** |
| | Blend | 97.85 | **97.98** | **98.90** | 98.62 | 98.39 | 98.04 | 97.86 | 98.75 | 98.47 | 98.34 |
| | CLB | 97.83 | 97.70 | 98.41 | 98.27 | 97.65 | 97.88 | **97.83** | **98.49** | 98.27 | **97.72** |
| | SIG | 97.50 | **97.44** | 98.56 | **98.21** | 98.05 | 97.88 | 97.36 | **98.67** | 98.14 | 97.89 |
| FT | BadNets | 93.79 | **94.29** | 96.64 | 96.09 | 95.82 | 94.03 | 94.16 | **96.86** | **96.66** | 95.52 |
| | Blend | 93.30 | **94.07** | 95.96 | **96.83** | **96.06** | 94.00 | 93.99 | **96.83** | 96.59 | 95.89 |
| | CLB | 94.06 | **94.28** | **96.67** | 96.39 | 95.53 | 94.20 | 94.01 | 96.24 | **96.50** | 95.92 |
| | SIG | **93.51** | **93.98** | 96.78 | 96.52 | 95.84 | 93.45 | 93.79 | **97.14** | 96.59 | 95.96 |
| FP | BadNets | 93.52 | 93.40 | 95.84 | 95.18 | **94.57** | **93.67** | **93.41** | **95.98** | **95.29** | 93.59 |
| | Blend | 92.59 | **94.06** | 95.94 | 94.69 | 94.37 | 93.05 | 93.96 | 96.11 | 95.43 | **94.79** |
| | CLB | **93.22** | 93.99 | 95.91 | 95.36 | **94.55** | 93.15 | **94.17** | 95.48 | 95.42 | 94.36 |
| | SIG | **93.88** | 93.36 | 95.97 | **95.50** | **94.54** | 93.75 | **93.84** | 96.24 | 95.20 | 94.37 |
| NAD | BadNets | 93.77 | **95.39** | 97.03 | **97.00** | 95.76 | **93.82** | 95.19 | **97.12** | 96.91 | **95.85** |
| | Blend | 94.09 | **95.85** | 97.12 | 96.77 | 95.93 | **94.12** | 95.57 | 97.08 | 96.51 | 95.92 |
| | CLB | 93.88 | **95.38** | 96.89 | 96.98 | 95.87 | 94.02 | 95.09 | 96.75 | 96.57 | **96.52** |
| | SIG | 93.86 | **95.51** | 97.20 | 96.95 | **96.23** | **93.95** | 95.22 | **97.52** | 96.95 | 95.62 |
| ANP | BadNets | 94.26 | 95.86 | **98.18** | 97.59 | 97.14 | **94.40** | **96.26** | 98.12 | 97.56 | 96.68 |
| | Blend | 92.70 | 96.47 | **98.18** | 98.00 | 97.14 | **95.67** | **96.70** | 98.14 | **98.47** | 96.68 |
| | CLB | 95.71 | 96.45 | 97.89 | 97.61 | 97.33 | **95.83** | **96.68** | **98.12** | **97.71** | 96.97 |
| | SIG | 92.60 | 96.55 | 97.87 | **97.73** | **97.91** | **94.62** | 96.55 | **98.01** | 97.69 | 97.47 |
| AWM | BadNets | **95.02** | 94.52 | **96.39** | 95.93 | **95.46** | 93.87 | **94.91** | 96.28 | **96.18** | 95.43 |
| | Blend | **95.08** | **94.99** | 93.00 | **96.51** | **96.00** | 95.06 | 94.82 | **95.38** | 96.28 | 94.40 |
| | CLB | **95.60** | **94.94** | **95.20** | 96.17 | 95.33 | 95.12 | 94.84 | 94.22 | **96.41** | **95.53** |
| | SIG | **94.58** | **94.76** | 96.89 | 96.59 | **96.05** | 94.46 | 94.43 | **96.90** | 96.57 | 95.80 |

SGD results in an average ACC drop of 24% in FP, much larger than 4.46% caused by AdamW. Besides, SGD also has a little worse defense performance.

## D   THE EFFECT OF FINE-TUNING EPOCHS ON FT, FP AND NAD

Table 8: The performance of Fine-tuning-based defenses for different fine-tuning epochs.

| Metric | Defense | epoch=20 | | | | epoch=100 | | | | |
|---|---|---|---|---|---|---|---|---|---|---|
| | | Badnets | Blend | CLB | SIG | Badnets | Blend | CLB | SIG | AvgDrop |
| ACC | FT | 93.79 | 93.30 | 94.06 | 93.51 | 90.30 | 90.43 | 91.20 | 90.19 | 3.14 |
| | FP | 93.52 | 92.59 | 93.22 | 93.88 | 89.86 | 90.01 | 89.56 | 89.45 | 3.58 |
| | NAD | 93.77 | 94.09 | 93.88 | 93.86 | 90.62 | 91.22 | 90.87 | 91.14 | 2.94 |
| ASR | FT | 2.51 | 4.91 | 1.33 | 1.40 | 1.26 | 3.15 | 1.48 | 0.93 | 0.83 |
| | FP | 0.91 | 0.73 | 1.70 | 0.81 | 1.08 | 1.01 | 2.13 | 0.80 | -0.22 |
| | NAD | 1.57 | 8.94 | 7.27 | 3.60 | 1.49 | 4.62 | 5.08 | 2.59 | 1.89 |

Here, we compare the performance of the fine-tuning-based methods for different fine-tuning epochs. As shown in Table 8, a notable accuracy drop appears on all defenses when we fine-tune the models for longer epochs, *e.g.*, the average accuracy drop is 3.14% in FT, which hinders the use of the model. With such a notable accuracy drop, ASR only decreases slightly, *e.g.*, 0.83% in FT with more epochs. Therefore, we recommend using fewer epochs to preserve the utility of the ViTs better.

## E   THE ACCURACY OF OUR ATTACK ON CIFAR-10 DATASET

We have discussed the attack performance of our proposed method as shown in Table 4 of Section 5.1. Here, we continue to explore the effect on the accuracy of our attacks. As shown in Table 9, the backdoored models with our method have comparable accuracy to their baselines (without our method), which indicates our method does not influence the utility of the backdoored model and guarantees the stealthiness of backdoored models with our method.

## F   THE SETTING OF OUR ATTACK ON IMAGENET DATASET

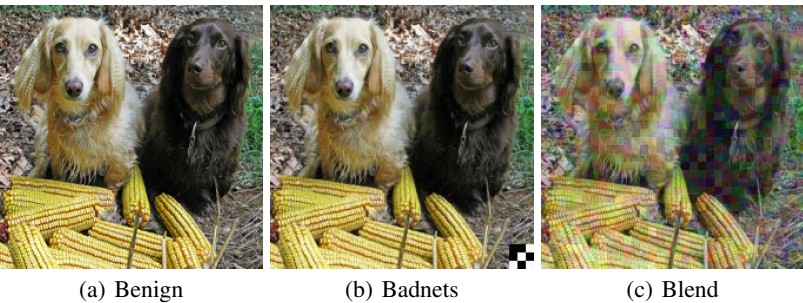

(a) Benign  (b) Badnets  (c) Blend

Figure 7: Examples for the benign and backdoor images on ImageNet dataset.

**Attack:** Since the huge computational cost, we fine-tune the pre-trained ViT-B on the poisoned ImageNet with 512 batch size and 10 epochs to insert backdoors. Because ImageNet is a high-resolution dataset, we increase the trigger size of badnets attacks to $21 \times 21$ for better poisoning. For the Blend attack, we resize the image of gaussian noise to $224 \times 224$ to accommodate the large input size on ImageNet. In Figure 7, we show examples of benign and backdoor images. For other settings of the vanilla poisoning, we keep the same with our experiments on CIFAR-10 (Please refer to Appendix A for details.). For the settings of our proposed attack, we follow the settings of CIFAR-10 except for the following two points: During the perturbation generation step, the budget and step size are set to $8/255$ and $2/255$, respectively. Similar to the vanilla backdoor attack, the patch size of RMP is enlarged to 16 because ImageNet is a high-resolution dataset. For ViT-specific attacks, we choose DeiT-B (Touvron et al., 2022) which has the exact same architecture as ViT-B for poisoning without any hyperparameter change.

**Defense:** First, for the defense methods unrelated to architectures, to achieve a better acceleration of the experiments on ImageNet, we adopt a large batch size of images for defense. In detail, for fine-tuning-based defense, the batch size is set to 512. For pruning-based defense, the batch size is set to 128 to avoid the out-of-memory problem on 4 NVIDIA 3090 GPUs. Other settings are the same as our experiment on CIFAR-10. Please refer to Appendix B for details. As for the ViT-specific attack: attention blocking (AB), we adopt the default setting recommended by (Subramanya et al., 2022b): during the inference stage, we block out the region of size $30 \times 30$ which is highlighted by Attention Rollout (Abnar & Zuidema, 2020).

## G   THE ACCURACY OF OUR ATTACK ON IMAGENET DATASET

Like the experiments on CIFAR-10, we also evaluate the effect of our method on ACC for large datasets like ImageNet. The results in Table 10 show that our method does not influence the utility of the backdoored models and the stealthiness of backdoored models on large datasets can also be further guaranteed.

Table 10: ACC (%) of our attack on ImageNet dataset. The higher ACC is in **bold**.

| Attack | Before | FT | FP | NAD | ANP | AWM | AB |
|---|---|---|---|---|---|---|---|
| TrojViT | 80.59 | 76.82 | 76.93 | 77.55 | 76.31 | 77.78 | - |
| DBIA | 79.52 | 78.3 | 75.2 | 77.18 | 76.49 | 78.94 | - |
| Badnets | 80.82 | 71.05 | 68.10 | 72.38 | 69.56 | 76.40 | **74.86** |
| CAT+Badnets | 81.01 | **71.41** | **68.31** | **72.69** | **69.79** | **76.62** | 74.51 |
| Blend | 80.82 | 71.03 | **68.43** | 72.60 | 69.69 | **76.77** | 74.72 |
| CAT+Blend | **81.03** | **71.12** | 68.39 | **72.62** | **69.96** | 76.36 | **74.73** |

## H   BROADER IMPACT

While our adaptation to backdoor defense eliminates backdoor behaviors inside backdoored ViTs, it is important to avoid creating overconfidence among readers regarding the robustness of current ViTs.

Note that there still may exist powerful attacks that can bypass these existing defenses, like the new attack we proposed in this paper. Furthermore, the proposed method is a strong attack to existing defense, thereby increasing potential risks in practical applications. However, we firmly believe that comprehensive evaluations using stronger attacks and more revealed potential risks would encourage practitioners to prioritize the security of their deployed models.

