# OpenReview forum: "Towards Reliable Backdoor Attacks on Vision Transformers"
_ICLR.cc/2024/Conference — Submitted to ICLR 2024_

### Official Review · Reviewer_Vd1v · 2023-10-20

**Soundness:** 2 fair
**Presentation:** 2 fair
**Contribution:** 2 fair
**Rating:** 5
**Confidence:** 4

**Summary:**

This paper examines the prevalent backdoor attacks and defenses, revealing an over-optimistic perception arising from the improper adaptation of defenses from CNNs to ViTs. With appropriate inheritance from CNNs, existing backdoor attacks can be effectively mitigated. Additionally, the paper introduces a more robust attack method: a minor perturbation on the trigger significantly enhances the resilience of existing attacks against diverse defenses.

**Strengths:**

It reveals an over-optimistic perception arising from the improper adaptation of defenses from CNNs to ViTs.

This paper introduces a more robust attack method against ViTs.

**Weaknesses:**

When testing existing backdoor attacks against ViT, the authors only use CNN-based backdoor attacks without ViT-specific backdoor attack methods. Thus, the possibility exists that existing ViT-specific backdoor attacks can also evade well-adapted backdoor defenses.

When testing existing backdoor defenses, the authors only consider purified-based backdoor defenses. How about the detection-based backdoor defenses? Are they also over-estimated?

Lack of enough baselines to prove the effectiveness of the proposed attack method. After proposing a new backdoor attack, the authors should compare it with SOTA backdoor attacks, especially advanced ViT-specific backdoor attacks, to show its superiority.

There is insufficient evaluation to explore whether the proposed attack can evade the SOTA backdoor defenses designed for ViT.

There is a lack of enough complex datasets, such as Imagenet, to evaluate the effectiveness of the proposed attacks.

**Questions:**

See the concerns in weakness.

---

> ### Author Response · Authors · 2023-11-20
> **Response to Reviewer Vd1v (Part1)**
>
> Dear reviewer Vd1v:
>
> Thank you for your valuable feedback. We are very sorry for the ambiguity in section 5.2. We have re-organized this section to make it easier to read.
>
> **Q1:**  Thus, the possibility exists that existing ViT-specific backdoor attacks can also evade well-adapted backdoor defenses.
>
> **A1:** TrojViT [1] and DBIA [2] are the existing two kinds of ViT-specific attacks. **In Table 5 and Table 10 in the paper, we have shown that both of them fail to evade well-adapted backdoor defenses**: they only achieve a very low ASR (<1%) across all model-agnostic defense approaches while the benign accuracy maintains a relatively high level (>75%). Here we further illuminate that our proposed adapted approaches remain essential for better defending TrojViT and DBIA. The experiments are performed on ImageNet. The ASR and ACC before and after adaptations are shown as follows:
>
> **ASR** (%)
>
> TrojViT:
>
> |                    | Before | FT   | FP   | NAD  | ANP  | AWM  |
> | ------------------ | ------ | ---- | ---- | ---- | ---- | ---- |
> | Before adaptations | 91.58  | 0.16 | 0.19 | 0.19 | 0.29 | 0.00 |
> | After adaptations  | 91.58  | 0.14 | 0.14 | 0.16 | 0.46 | 0.18 |
>
> DBIA:
>
> |                    | Before | FT   | FP   | NAD  | ANP  | AWM  |
> | ------------------ | ------ | ---- | ---- | ---- | ---- | ---- |
> | Before adaptations | 99.58  | 0.07 | 0.10 | 0.24 | 0.13 | 0.06 |
> | After adaptations  | 99.58  | 0.09 | 0.07 | 0.10 | 0.10 | 0.05 |
>
> **ACC** (%)
>
> TrojViT:
>
> |                    | Before | FT        | FP        | NAD       | ANP       | AWM       |
> | ------------------ | ------ | --------- | --------- | --------- | --------- | --------- |
> | Before adaptations | 80.59  | 68.92     | 69.00     | 69.88     | 69.72     | 0.10      |
> | After adaptations  | 80.59  | **76.82** | **76.93** | **77.55** | **76.31** | **77.18** |
>
> DBIA:
>
> |                    | Before | FT        | FP        | NAD       | ANP       | AWM       |
> | ------------------ | ------ | --------- | --------- | --------- | --------- | --------- |
> | Before adaptations | 79.52  | 69.36     | 64.05     | 67.51     | 71.04     | 57.00     |
> | After adaptations  | 79.52  | **78.30** | **75.20** | **77.18** | **76.49** | **78.94** |
>
> The results demonstrate that although vanilla settings can successfully decrease the ASR, they also impair the benign function of the backdoored models a lot. In contrast, our proposed adaptations better preserve the benign ACC. For example,  AWM achieves the ACC of 77.18% after adaptations against TrojViT while it totally collapses without any adaptations.
>
> **Q2:**  The authors only consider purified-based backdoor defenses. How about the detection-based backdoor defenses? Are they also over-estimated?
>
> **A2:** Actually,  the detection-based defenses are underestimated for current attacks, but our proposed CAT can better bypass those defenses. Take NC [3] as an example, which is one of the most popular detection-based backdoor defenses. It is composed of two stages: It first searches the possible trigger through outlier detection. Then, the backdoor model is mitigated through unlearning with the reversed trigger. Performing experiments on the CIFAR-10 dataset for ViT-B, we explore the performance of NC by dividing the two stages separately:
>
> **Stage 1: Detection**
>
> NC reconstructs potential triggers for each class and uses an anomaly index to determine if one of them is a valid trigger. The larger the anomaly index, the more likely it is to be a real backdoor trigger. Here, we calculate the anomaly indexes of the attack with or without CAT for comparison:
>
> |         | Badnets | Blend | CLB  | SIG  |
> | ------- | ------- | ----- | ---- | ---- |
> | Vanilla | 7.45    | 3.14  | 7.13 | 2.26 |
> | +CAT    | **5.04**    | **1.60**  | **2.48** | **0.90** |
>
> The results show that CAT can always obtain lower anomaly indexes: for example, the vanilla badnets attack obtains the anomaly indexes of 7.45 which is quite larger than those after combing CAT (5.04). It means CAT can help existing attacks better bypass the detection of NC.
>
> **Stage 2: Unlearning**
>
> Next, the defenders use the reconstructed triggers to mitigate the backdoor behavior once the reconstructed triggers are identified. Specifically, they fine-tune the model to predict ground-truth labels in the presence of the triggers, i.e., unlearning the backdoor behavior. Here, we explore whether CAT makes existing attacks more resistant to unlearning. According to previous research [1] which observes that the unlearning process of NC with CNNs’ default settings will decrease the benign acc a lot (~50%), we make the following adaptations based on the observations in our paper:
>
> - Use AdamW optimizer to unlearn the backdoored models.
> - Unlearn the backdoored model only for 20 epochs.

---

> ### Author Response · Authors · 2023-11-20
> **Response to Reviewer Vd1v (Part2)**
>
> We summarize the results as follows:
>
> ASR (%)
>
> |         | Badnets   | Blend     | CLB      | SIG       |
> | ------- | --------- | --------- | -------- | --------- |
> | Vanilla | 1.08      | 0.66      | 0.36     | 5.64      |
> | CAT     | **99.99** | **53.49** | **6.25** | **43.79** |
>
> ACC (%)
>
> |         | Badnets | Blend | CLB   | SIG   |
> | ------- | ------- | ----- | ----- | ----- |
> | Vanilla | 96.85   | 96.61 | 96.78 | 96.78 |
> | CAT     | 97.22   | 97.08 | 96.75 | 97.06 |
>
> We first observe that NC successfully mitigates the effect of all existing attacks and maintains high ACC.  Secondly, CAT can help current attacks bypass the unlearning process better: for example, the ASR of Badnets is improved from 1.08% to 99.99%.
>
> **Q3:** After proposing a new backdoor attack, the authors should compare it with SOTA backdoor attacks, especially advanced ViT-specific backdoor attacks, to show its superiority.
>
> **A3:**  **In fact, we have compared CAT with two ViT-specific attacks: TrojViT [1] and DBIA [2] in Table 5 and 10 in the paper.**  We summarize the average ASR of all attack methods after performing five model-agnostic defenses in the following table.
>
> |                | DBIA | TrojViT | badnets+CAT | blend+CAT |
> | -------------- | ---- | ------- | ----------- | --------- |
> | Average ASR(%) | 0.08 | 0.21    | 44.12       | 32.84     |
>
> The results demonstrate that current ViT-specific attacks can not be immune to the adapted defenses and  CAT can outperform them with a higher ASR.
>
> **Q4:** There is insufficient evaluation to explore whether the proposed attack can evade the SOTA backdoor defenses designed for ViT.
>
> **A4:** **In Table 5 and Table 10 in the paper, we have shown that CAT can evade the AB (Attention block) [5] which is one of the ViT-specific backdoor defenses.** Here we further perform experiments on patch-processing [6] which is another ViT-specific defense.  Following their original paper, we select TPR (the proportion of benign samples that are successfully identified) and TNR (the proportion of backdoor samples that are successfully detected) as our metrics. The experiments are performed on ImageNet with ViT-B architecture.
>
> |           | TPR   | TNR   |
> | --------- | ----- | ----- |
> | Blend     | 80.12 | 13.03 |
> | Blend+CAT | 81.09 | **5.15**  |
>
> Compared to the model-agnostic defense in Table 5, patch-processing only obtains limited performance on both Blend and Blend+CAT. We speculate this is because ImageNet is a large and complex dataset. It makes the classification results more susceptible to change if certain patches of the image are masked, thereby increasing the difficulty of distinguishing between clean and backdoor images. In addition, combining with CAT helps Blend obtain lower TNR, representing the backdoor samples are harder to detect. We also perform experiments on CIFAR-10 to further investigate whether CAT can provide better robustness on small datasets. The experiments are also performed on ViT-B:
>
> |           | TPR   | TNR   |
> | --------- | ----- | ----- |
> | Blend     | 83.08 | 97.11 |
> | Blend+CAT | 84.11 | **48.69** |
>
> Similar to the experiments on Imagenet, combining with CAT can obtain comparable TPR and lower TNR (<50%). This demonstrates that CAT can evade the SOTA backdoor defenses designed for ViT.
>
> **Q5:** There is a lack of enough complex datasets, such as ImageNet, to evaluate the effectiveness of the proposed attacks.
>
> **A5:** **We actually perform experiments on ImageNet in Section 5.2 in the paper.** The results show that our proposed attacks are still effective on large datasets.
>
> We really hope our explanation can eliminate your doubts. We are happy for the further discussion with you if any concerns remain.
>
> [1] TrojViT: Trojan Insertion in Vision Transformers, Mengxin Zheng et al., In CVPR 2023.
>
> [2] Dbia: Data-free backdoor injection attack against transformer networks, Peizhuo Lv et al., In Arxiv 2021.
>
> [3] Neural Cleanse: Identifying and Mitigating Backdoor Attacks in Neural Networks, Bolun Wang et al., In S&P 2019.
>
> [4] BackdoorBench: A Comprehensive Benchmark of Backdoor Learning, Baoyuan Wu et al., In NeurIPS 2022.
>
> [5] Backdoor attacks on vision transformers. Akshayvarun Subramanya et al., In Arxiv 2022.
>
> [6] Defending Backdoor Attacks on Vision Transformer via Patch Processing, Khoa D. Doan et al., in AAAI 2023.

---

> > ### Author Response · Authors · 2023-11-22
> > **Look forward to your reply**
> >
> > Dear Reviewer Vd1v,
> >
> >  We would like to express our sincere gratitude to your insightful review. The experiments in our paper and our additional experiments demonstrate that:
> >
> > - Even the state-of-the-art ViT-specific attacks fail to evade well-adapted backdoor defenses. Adaptation is still needed to maintain high ACC.
> > - The detection-based backdoor defenses still need adaptations. CAT is capable of bypassing those defenses.
> > - In the table 5 and table 10 of our paper, we have demonstrated that CAT outperforms advanced ViT-specific attacks in a novel margin.
> > - CAT can envade the SOTA ViT-specific defenses like AB and patch-processing.
> > - In the table 5 and table 10 of our paper, we have demonstrated that CAT is effectiveness on ImageNet dataset.
> >
> > As the final deadline of reviewer-author discussion is approaching, we still look forward to your valuable feedback if any concerns remain. We are ready to provide further elaboration and engage in a more in-depth discussion. If you are satisfied with our replies, please don't hesitate to update your score.
> >
> > Best wishes,
> >
> > Authors

---

### Official Review · Reviewer_FbZc · 2023-11-01

**Soundness:** 2 fair
**Presentation:** 3 good
**Contribution:** 2 fair
**Rating:** 5
**Confidence:** 4

**Summary:**

This paper first conducts a comprehensive evaluation of existing backdoor attacks on ViT and reveals the reason they can bypass existing defense is due to the inappropriate use of optimizer, e.g., SGD. After refining the existing backdoor defense, the experiment results show that existing backdoor attacks on ViT will no longer achieve effective attack after defense. Therefore, the authors propose a more reliable attack by adding special adversarial perturbations into the trigger pattern. The results show their method can achieve a stable attack after some type of defense.

**Strengths:**

The authors revisit the existing backdoor defense methods on ViT and find that these defense methods don’t work well because of the misuse of the optimizer, i.e. SGD.

The authors conduct comprehensive experiments and ablation studies.

**Weaknesses:**

The hypothesis lacks enough evidence. Firstly, the authors claim “ViTs are typically trained by AdamW while its fine-tuning defense is trained by SGD (NOT AdamW, maybe inheriting from CNNs). This discrepancy in optimizers raises the possibility that the perceived vulnerability of ViTs (with defense) might be overstated, i.e., the success of attacks on ViTs with defense may be questionable.” However, the authors don’t cite papers that use SGD to mitigate backdoors in ViT. And when transferring the defense methods on CNN to ViT, the most straightforward scheme is to use the same optimizer as when training the model, i.e., SGD for CNN and AdamW for ViT. Secondly, the authors claim that the misuse of the optimizer in fine-tuning leads to suboptimal defense performance and conduct experiments in Table 2 to show the effect of optimizers. However, the attack methods used in Table 2 are all CNN-specific attack methods. Authors should conduct experiments on ViT-specific backdoor attacks [2,3,4] because they are investigating backdoor defense on ViTs.


The “backdoor defense” in the paper only denotes the “mitigation” aspect. And the design of their reliable attack is based on “the difference in the intermediate-level representations between the inputs with and without triggers”. It is not clear if this attack can bypass detection technologies that don’t rely on the difference in activation, such as Neural Cleanse[1] which is based on reverse engineering and outlier detection.

**Questions:**

Is the proposed attack only effective on ViT? Is it possible that it also works well on CNN, since the proposed method doesn’t leverage ViT’s unique features compared to CNN?

Same with weakness 2, is it possible that the authors can provide results of the attacks against backdoor detection techniques such as Neural Cleanse [1]?

[1] B. Wang et al., "Neural Cleanse: Identifying and Mitigating Backdoor Attacks in Neural Networks," 2019 IEEE Symposium on Security and Privacy (SP), San Francisco, CA, USA, 2019, pp. 707-723, doi: 10.1109/SP.2019.00031.

[2] Zheng, Mengxin, Qian Lou, and Lei Jiang. "Trojvit: Trojan insertion in vision transformers." Proceedings of the IEEE/CVF Conference on Computer Vision and Pattern Recognition. 2023.

[3] Zheng, Runkai, et al. "Data-free backdoor removal based on channel lipschitzness." European Conference on Computer Vision. Cham: Springer Nature Switzerland, 2022.

[4] Akshayvarun Subramanya, Aniruddha Saha, Soroush Abbasi Koohpayegani, Ajinkya Tejankar, and Hamed Pirsiavash. Backdoor attacks on vision transformers. arXiv preprint arXiv:2206.08477, 2022a.

---

> ### Author Response · Authors · 2023-11-20
> **Response to Reviewer FbZc (Part1)**
>
> Dear reviewer FbZc:
>
> Thanks very much for your constructive and detailed comments. Here are our responses to address your concerns:
>
> **Q1:**  The hypothesis lacks enough evidence. However, the authors don’t cite papers that use SGD to mitigate backdoors in ViT. When transferring the defense methods on CNN to ViT, the most straightforward scheme is to use the same optimizer as when training the model, i.e., SGD for CNN and AdamW for ViT.
>
> **A1:** We apologize for the confusion in Section 3.2. In fact, we cited previous research [1], which used the SGD for defensive fine-tuning while Adam for pretraining. To avoid misunderstanding, we revised our paper and discussed previous research in section 3.2 on page 4. Note that the earliest work [2], which first introduced transformers to computer vision, used AdamW for pre-training and SGD for fine-tuning. Following studies [3, 4, 5] naturally inherit this strategy. Unfortunately, in backdoor defense, we found that this natural choice caused issues. We want to use this article to remind everyone to pay attention to making reasonable choices rather than blindly adopting default or natural options.
>
> **Q2:** The attack methods used in Table 2 are all CNN-specific attack methods. Authors should conduct experiments on ViT-specific backdoor attacks [6,7,8].
>
> **A2:** Thank you for your suggestion. First, we argue that these backdoor attacks in Table 2 are actually model-agnostic and general rather than CNN-specific. Second, we conduct experiments on ViT-specific attacks to make our conclusion more convincing. Here, we consider the effect of the optimizer on FT for TrojViT [6] and DBIA attacks [7]. We do not include previous research [8] since it proposed to use attention to alleviate the backdoor effect for defense.
> The results are shown in the following table. The model is DeiT-B, and all experiments are performed on ImageNet. It is easily found that for ViT-specific attacks, AdamW is also a better choice, which better preserves the ACC and reduces the ASR to an extremely low level. This phenomenon is the same as those with model-agnostic attacks in Table 2.
>
> ASR (%)
>
> |            | TrojViT  | DBIA     |
> | ---------- | -------- | -------- |
> | No defense | 91.08    | 99.58    |
> | SGD        | 0.37     | 0.12     |
> | AdamW      | **0.14** | **0.09** |
>
> ACC (%)
>
> |            | TrojViT   | DBIA      |
> | ---------- | --------- | --------- |
> | No defense | 80.59     | 79.52     |
> | SGD        | 69.63     | 70.26     |
> | AdamW      | **76.82** | **78.30** |
>
> **Q3:** It is not clear if this attack can bypass detection technologies that don’t rely on the difference in activation, such as Neural Cleanse [9] which is based on reverse engineering and outlier detection.
>
> **A3:** Here, we use Neural Cleanse (NC) as an example. NC is a backdoor consisting of two steps: (1) It first reconstructs all possible triggers through optimization and determines whether a backdoor attack exists via outlier detection. (2) it mitigates backdoor behavior through unlearning with the reconstructed trigger, i.e., restoring the performance even with the presence of the trigger. We examine whether CAT can better bypass NC in these two stages, and all experiments are performed on the CIFAR-10 dataset under ViT-B.
>
> **Stage 1: Detection**
>
> NC reconstructs potential triggers for each class and uses an anomaly index to determine if one of them is a valid trigger. The larger the anomaly index, the more likely it is to be a real backdoor trigger. Here, we calculate the anomaly indexes of the attack with or without CAT for comparison,
>
> |         | Badnets | Blend | CLB  | SIG  |
> | ------- | ------- | ----- | ---- | ---- |
> | Vanilla | 7.45    | 3.14  | 7.13 | 2.26 |
> | +CAT    | 5.04    | 1.60  | 2.48 | 0.90 |
>
> The results show that CAT can always lower anomaly indexes, making the attack stealthier. For example, the vanilla badnets attack obtains anomaly indexes of 7.45, which is larger than those after combing CAT (5.04). It means CAT can help existing attacks better bypass the detection of NC.

---

> > ### Author Response · Authors · 2023-11-20
> > **Response to Reviewer FbZc (Part2)**
> >
> > **Stage 2: Unlearning**
> >
> > Next, the defenders use the reconstructed triggers to mitigate the backdoor behavior once the reconstructed triggers are identified. Specifically, they fine-tune the model to predict ground-truth labels in the presence of the triggers, i.e., unlearning the backdoor behavior. Here, we explore whether CAT makes existing attacks more resistant to unlearning. According to previous research [1] which observes that the unlearning process of NC with CNNs’ default settings will decrease the benign acc a lot (~50%), we make the following adaptations based on the observations in our paper:
> >
> > - Use AdamW optimizer to unlearn the backdoored models.
> > - Unlearn the backdoored model only for 20 epochs.
> >
> > We summarize the results as follows:
> >
> > ASR (%)
> >
> > |         | Badnets | Blend | CLB  | SIG   |
> > | ------- | ------- | ----- | ---- | ----- |
> > | Vanilla | 1.08    | 0.66  | 0.36 | 5.64  |
> > | CAT     | **99.99**   | **53.49** | **6.25** | **43.79** |
> >
> > ACC (%)
> >
> > |         | Badnets | Blend | CLB   | SIG   |
> > | ------- | ------- | ----- | ----- | ----- |
> > | Vanilla | 96.85   | 96.61 | 96.78 | 96.78 |
> > | CAT     | 97.22   | 97.08 | 96.75 | 97.06 |
> >
> > The table shows that CAT can make unlearning more difficult and keeps backdoor behavior inside the model.
> >
> > In conclusion, CAT helps us escape both Anomaly Index-based detection and the subsequent unlearning process.
> >
> > **Q4:** Is the proposed attack only effective on ViT? Is it possible that it also works well on CNN, since the proposed method doesn’t leverage ViT’s unique features compared to CNN?
> >
> > **A4:** Directly using our poisoned datasets crafted by CAT for training, we perform experiments on CIFAR-10 with ResNet50. Here, we summarize the ASR after performing FT and AWM against Badnets attack in the following tables:
> >
> > |      | Vanilla | CAT   |
> > | ---- | ------- | ----- |
> > | FT   | 27.69   | **48.00** |
> > | AWM  | 0.93    | **4.81**  |
> >
> > Surprisingly, for CNNs, CAT can still improve ASR for defending both the fine-tuning-based defense and pruning-based defense. This demonstrates that CAT may offer a reliable evaluation of backdoor robustness for other architectures as well.
> >
> > Hope our responses can clear up your doubts and answer your questions. Look forward to your replies!
> >
> > [1] BackdoorBench: A Comprehensive Benchmark of Backdoor Learning, Baoyuan Wu et al., In NeurIPS 2022.
> >
> > [2] An Image is Worth 16$\times$16 Words: Transformer for Image Recognition at Scale, Dosovitskiy et al, In ICLR 2021.
> >
> > [3] How to train your ViT? Data, Augmentation, and Regularization in Vision Transformers, Steiner et al., In Transactions on Machine Learning Research 2022.
> >
> > [4] Tokens-to-Token ViT: Training Vision Transformers from Scratch on ImageNet, Yuan et al., in ICCV 2021.
> >
> > [5] CvT: Introducing Convolutions to Vision Transformers, Wu et al, in ICCV 2021.
> >
> > [6] TrojViT: Trojan Insertion in Vision Transformers, Mengxin Zheng et al., In CVPR 2023.
> >
> > [7] Dbia: Data-free backdoor injection attack against transformer networks, Peizhuo Lv et al., In Arxiv 2021.
> >
> > [8] Backdoor attacks on vision transformers. Akshayvarun Subramanya et al., In Arxiv 2022.
> >
> > [9] Neural Cleanse: Identifying and Mitigating Backdoor Attacks in Neural Networks, Bolun Wang et al., In S&P 2019.

---

> > > ### Author Response · Authors · 2023-11-22
> > > **Look forward to your reply**
> > >
> > > Dear Reviewer FbZc,
> > >
> > >  We would like to express our sincere gratitude to your insightful review. Hope our responses have successfully eliminate your doubts regarding the evidence of our hypothesis, the comparison of optimizers on ViT-specific attacks. In addition, our additional experiments support that CAT is still effective in better bypassing NC with lower anomaly indexes and higher ASR after unlearning. After peforming experiments on ResNet50, we superisingly find that CAT might also potentially provide reliable evaluatiion of backdoor robustness for other architectures. As the final deadline of reviewer-author discussion is approaching, we still look forward to your valuable feedback if any concerns remain. We are ready to provide further elaboration and engage in a more in-depth discussion. If you are satisfied with our replies, please don't hesitate to update your score.
> > >
> > > Best wishes,
> > >
> > > Authors

---

### Official Review · Reviewer_jtUw · 2023-11-01

**Soundness:** 2 fair
**Presentation:** 2 fair
**Contribution:** 2 fair
**Rating:** 5
**Confidence:** 3

**Summary:**

This paper study backdoor attack on Vision Transformers. They show that existing defenses successfully defend against backdoor attacks in ViT-B and CIFAR10 dataset. Moreover, they proposed Channel Activation attack (CAT). They show that CAT attack is more effective on CIFAR10 dataset.

**Strengths:**

[+] CAT attack can transfer to other Vision transformers on Table 4 in CIFAR10 dataset.

**Weaknesses:**

[-] Study of backdoor attack with only CIFAR10 and single vision transformer architecture is not convincing, and any conclusion based on these limited settings won’t be accurate. Note that study of backdoor attack on the Vision Transformer has been conducted before.

[-] What is your thread model in your proposed attack? Do you assume that adversary have access to the model during training? Current setting is confusing to me since there are two thread models: 1. Both source and target being same model 2. Source and target are different


[-] In ImageNet experiments, both source and target are ViT-B. Does this means that adversary has access to model architecture and its parameters. This is not a practical scenario in my opinion and limits the impact of the paper.

**Questions:**

-

---

> ### Author Response · Authors · 2023-11-20
> **Response to Reviewer jtUw (Part1)**
>
> Dear reviewer jtUw:
>
> Thank you for your invaluable reviews. For your concerns, our replies are as follows:
>
> **Q1:** The study of backdoor attacks with only CIFAR10 and single vision transformer architecture is not convincing, and any conclusion based on these limited settings won’t be accurate. Note that study of backdoor attack on the Vision Transformer has been conducted before.
>
> **A1:**  Firstly, we would like to reclaim our contributions compared to previous works [1-4] studying the backdoor robustness for ViTs:
>
> - **Backdoor Attack:** Zheng et al. in [1] and Lv et al. in [2] propose two ViT-specific attacks respectively: TrojViT and DBIA. However, our works show that both of them fail to evade the backdoor defenses after adaptations. **Only our proposed CAT can maintain a high ASR.**
> - **Backdoor Defense:** In [3], they first apply model-agnostic backdoor defense on ViTs. However, due to the lack of any adaptation, they either significantly reduce ACC or are unable to reduce ASR. Therefore, they conclude that: the effective defense that has been verified on the CNN architecture may not be suitable for the ViT architecture. In contrast to their conclusions, **we find that with proper adaptation, current defense methods are still demonstrated to be effective on ViTs.** Subramanya et al. in [4] propose a ViT-specific defense: AB (Attention Blocking). We show that **CAT can improve the ASR even for ViT-specific defense like AB.**
>
> Furthermore, to illustrate that our conclusion is general and convincing, we conduct experiments on the CIFAR-100 dataset with the ViT-B and Swin-B architectures as follows:
>
> **Comparison of Optimizers**
>
> We perform FT using SGD and AdamW optimizer, respectively in the following tables:
>
> ViT-B
>
> |        | Before | SGD   | AdamW     |
> | ------ | ------ | ----- | --------- |
> | ACC(%) | 89.54  | 42.76 | **69.12** |
> | ASR(%) | 100.00 | 1.37  | 2.31      |
>
> Swin-B
>
> |        | Before | SGD  | AdamW     |
> | ------ | ------ | ---- | --------- |
> | ACC(%) | 92.24  | 2.81 | **78.56** |
> | ASR(%) | 100    | 0    | 1.39      |
>
> While the SGD optimizer can reduce the ASR to almost zero, it will significantly sacrifice the benign ACC, which makes the purified models unusable. By contrast, the ACC for fine-tuning using AdamW (69.12%) is much higher than the ACC for fine-tuning using SGD (42.76%). This observation is consistent with our findings on the CIFAR-10 dataset.
>
> **Comparison of Fine-tuning Epochs**
>
> On CIFAR-100, we also fine-tuned ViT-B and Swin-B with various epochs to investigate the effect of longer training on ViTs’ performance. The benign accuracy is shown in the following table,
>
> |        | Before | 20 epoch  | 100 epoch |
> | ------ | ------ | --------- | --------- |
> | ViT-B  | 89.54  | **69.12** | 58.93     |
> | Swin-B | 92.24  | **78.56** | 72.66     |
>
> where we find the severe overfitting issue caused by longer training. For example, the ACC drop of 100 epochs with ViT-B is more than 10% (69.12 → 58.93). This is consistent with the phenomenon in our paper.
>
> **Comparison of Puning Granularity**
>
> Taking AWM as an example, we investigate the impact of pruning granularity on the performance of defense approaches. Here, we compare the default granularity and our adapted granularity on CIFAR-100 with both ViT-B and Swin-B in the following tables.
>
> ViT-B
>
> |        | Before | Vanilla | Adapted   |
> | ------ | ------ | ------- | --------- |
> | ACC(%) | 89.54  | 61.04   | **82.07** |
> | ASR(%) | 100.00 | 1.17    | 1.18      |
>
> Swin-B
>
> |        | Before | Vanilla | Adapted   |
> | ------ | ------ | ------- | --------- |
> | ACC(%) | 92.24  | 39.97   | **81.81** |
> | ASR(%) | 100.00 | 0.41    | 2.60      |
>
> Similarly, our proposed coarse granularity (pruning the output channel of the linear projection layer) surpasses the default granularity (pruning ViTs using an element-wise mask) in preserving ACC. For example, our proposed adaptation improves ACC of ViT-B by a notable margin (61.04% → 82.07%) while keeping the ASR almost unchanged. The results demonstrate that pruning the output channel is a better choice across datasets or architectures.

---

> > ### Author Response · Authors · 2023-11-20
> > **Response to Reviewer jtUw (Part2)**
> >
> > **Q2:** What is your threat model in your proposed attack? Do you assume that the adversary has access to the model during training?
> >
> > **A2:** Sorry for the ambiguity in Section 5. We have revised the paper to make it clearer. Our primary focus is on the most common threat model, a **black-box scenario** (the source model is ViT-B, while the target model is another) where the attacker only knows the training data but is unaware of the training process and model. However, due to the difficulty of the black-box scenario, we initially considered a simpler **white-box scenario** (the source and target model are both ViT-B), where the attacker is aware of both the training data and the pre-trained model used for backdoor training. It's important to note that we have no access to the victim model to generate poisoned data since this victim model is fine-tuned from the pre-trained model based on already generated poisoned data.
> > Therefore, this paper considers two threat models in experiments: 1. **white-box scenario**: both source and target have the same architecture and backdoor training from the same pre-trained model; 2. **black-box scenario**: source and target are different. Among these, the second threat model is our primary focus.
> >
> > **Q3:**  In ImageNet experiments, both source and target are ViT-B. Does this mean that the adversary has access to model architecture and its parameters? This is not a practical scenario.
> >
> > **A3:** As mentioned in A2, the white-box scenario serves as a simpler and transitional phase in our research, while the more complex black-box scenario is our actual objective. We admit that performance on the white-box scenario is insufficient to demonstrate the efficacy of our approach with ViT. To address this, we supplement the results in the black-box scenario, where the source model is ViT-B, while the target model is Swin-B. The ACC and ASR  on ImageNet with or without CAT are summarized as follows:
> >
> > ASR (%):
> >
> > |             | Before | FT        | FP        | NAD       | ANP       | AWM       |
> > | ----------- | ------ | --------- | --------- | --------- | --------- | --------- |
> > | Badnets     | 100    | 31.91     | 22.23     | 42.36     | 42.29     | 35.92     |
> > | Badnets+CAT | 100    | **79.17** | **35.96** | **61.18** | **73.62** | **59.58** |
> > | Blend       | 100    | 6.10      | 3.01      | 18.00     | 10.85     | 31.96     |
> > | Blend+CAT   | 100    | **21.62** | **22.32** | **35.11** | **43.84** | **45.92** |
> >
> > ACC (%):
> >
> > |             | Before | FT    | FP    | NAD   | ANP   | AWM   |
> > | ----------- | ------ | ----- | ----- | ----- | ----- | ----- |
> > | Badnets     | 83.02  | 77.28 | 76.06 | 77.45 | 68.93 | 75.78 |
> > | Badnets+CAT | 83.11  | 77.27 | 76.10 | 77.86 | 69.54 | 76.84 |
> > | Blend       | 82.93  | 76.70 | 76.37 | 77.81 | 68.23 | 76.22 |
> > | Blend+CAT   | 83.09  | 76.87 | 76.51 | 77.26 | 70.16 | 75.96 |
> >
> > Similar to the results in Table 5, CAT brings negligible impact on ACC. Meanwhile, CAT improves ASR by a notable margin. For example, after combining CAT, the ASR for FP increases from 22.23% to 35.96% (+13.73%) against Badnets attack. This demonstrates that our proposed method still works on the black-box scenario.
> >
> > Hope our replies successfully addresss your concerns. Looking forward to the further discussion if other concerns remain.
> >
> > [1] TrojViT: Trojan Insertion in Vision Transformers, Mengxin Zheng et al., In CVPR 2023.
> >
> > [2] Dbia: Data-free backdoor injection attack against transformer networks, Peizhuo Lv et al., In Arxiv 2021.
> >
> > [3] BackdoorBench: A Comprehensive Benchmark of Backdoor Learning, Baoyuan Wu et al., In NeurIPS 2022.
> >
> > [4] Backdoor attacks on vision transformers. Akshayvarun Subramanya et al., In Arxiv 2022.

---

> > > ### Author Response · Authors · 2023-11-22
> > > **Look forward to your reply**
> > >
> > > Dear Reviewer jtUw,
> > >
> > >  We would like to express our sincere gratitude to your insightful review. Hope our responses have successfully eliminate your doubts regarding the reliable of our findings, the necessity of our work and the threat model in our proposed attack. In addition, our additional experiments demonstrate that CAT is still effective in the black-box scenario on ImageNet dataset. As the final deadline of reviewer-author discussion is approaching, we still look forward to your valuable feedback if any concerns remain. We are ready to provide further elaboration and engage in a more in-depth discussion. If you are satisfied with our replies, please don't hesitate to update your score.
> > >
> > > Best wishes,
> > >
> > > Authors

---

### Official Review · Reviewer_FJ73 · 2023-11-05

**Soundness:** 2 fair
**Presentation:** 2 fair
**Contribution:** 2 fair
**Rating:** 3
**Confidence:** 3

**Summary:**

The article identifies shortcomings in existing defense methods against backdoor attacks on Neural Networks, specifically focusing on Vision Transformers (ViT). It highlights deficiencies in fine-tuning-based defense and pruning-based defense on ViT and proposes adjustments to enhance their performance. Additionally, the authors introduce a new backdoor attack method, CAT, designed to bypass these defenses with increased robustness. CAT involves adding special adversarial perturbations to the trigger pattern to minimize noticeable channel activation differences between benign and triggered input.

**Strengths:**

- This paper is easy to understand.
- The article observes the use of different optimizers for training Convolutional Neural Networks (CNNs) and ViTs, suggesting a potential overstatement of ViTs' vulnerability to attacks with defense.
- The CAT attack seems effective in attacking the ViT models.

**Weaknesses:**

- The contributions of this paper seem incremental, especially in the defense part. The experiments indicate that optimizing the choice of optimizer, adjusting epoch numbers, and selecting appropriate granularity for pruning can improve defense performance on ViT. To apply fine-tuning-based methods to ViT, the authors adjust optimizers and epochs. However, these improvements are based on experimental trials, and there is no methodology to guide us on how to pick good hyperparameters.
- In Section 3.2, the impact of the epoch on fine-tuning defense is explored. The curve for the first 20 epochs differs significantly from the first 20 epochs when setting the experiment to 100 epochs, particularly in the left plot of (a) left. The variability in experimental results raises concerns about the reliability of the findings, considering the potential instability.
- Table 4 illustrates that the CAT attack method improves ASR, but the enhancement is limited, as most unsuccessful attacks do not become successful.
- Some symbols used in the formulas lack explanations. Appendix C Figure 7 should refer to Table 7.

**Questions:**

See my comments above.

---

> ### Author Response · Authors · 2023-11-20
> **Response to Reviewer FJ73 (Part1)**
>
> Dear reviewer FJ73:
>
> First, I would like to express my gratitude to you for your insightful feedback. For your raised questions, here are our responses:
>
> **Q1:** The contributions of this paper seem incremental, especially in the defense part. There is no methodology to guide us on how to pick good hyperparameters.
>
> **A1:** We believe that when solving problems, the greatest contribution to the community is to explore the fundamental issues and solve them with straightforward strategies, rather than a seemingly novel and complex approach. Here, we attribute the failure of the current defense to the following two points:
>
> - **The mismatch between training and defense**: In fact, the first paper proposing ViTs [1] uses AdamW for pretraining and finetuning with SGD. Therefore, multiple works such as [2] adopt this strategy as default and they only obtain 42.00% ACC. We first identify this problem and propose that the AdamW optimizer is essential for fine-tuning-based defense.
> - **The overfitting problem during the defense:** There is a typical problem when only a small amount of data is available for defense. In this situation, fine-tuning a large-capacity model is risky as it can easily lead to overfitting, especially for ViTs which are known to lack inductive bias [1]. Therefore, applying coarser granularity to reduce the number of learnable parameters and decreasing the number of iterations to lower the actual exploration parameter space can both reduce overfitting issues.
>
> Therefore, we believe that our approaches provide the community with sufficient insight and contributions. From our investigations, there are three methodologies that can be concluded to guide us in picking good hyperparameters.
>
> - If the defense methods are based on fine-tuning, the AdamW optimizer is recommended.
> - If the defense methods are based on pruning, pruning the channel of the linear layer might be a good choice.
> - For all defense methods, additional measures are needed to be taken to prevent overfitting.
>
> **Q2:** The curve for the first 20 epochs differs significantly from the first 20 epochs when setting the experiment to 100 epochs, particularly in the left plot of (a) left. The variability in experimental results raises concerns about the reliability of the findings, considering the potential instability.
>
> **A2**: This is because we employ a cosine learning rate (lr) schedule, which starts with a very large learning rate (0.0003) and gradually decreases until it reaches 0 in the final epoch. lr decreases at a different pace between training for 20 epochs and training for 100 epochs. Specifically, in the former case, lr drops to 0 at the 20th epoch (20 epochs in total), whereas in the latter case, it still keeps a large value (0.000274) at the 20th epoch (100 epochs in total). As a result, the curve seems quite different for the first 20 epochs.
>
> In addition, we repeated our experiments 3 times with various numbers of epochs in the following table, in which we can find our method is stable with small variability and the findings are reliable.
>
> **Badnets Attack**
>
> | fine-tuning epoch | 20          | 40          | 60          | 80          | 100         |
> | ----------------- | ----------- | ----------- | ----------- | ----------- | ----------- |
> | ACC               | 93.71 $\pm$ 0.14 | 92.42 $\pm$ 0.10 | 92.23 $\pm$ 0.14 | 91.25 $\pm$ 0.10 | 90.48 $\pm$ 0.17 |
> | ASR               | 2.08 $\pm$ 0.60  | 1.92 $\pm$ 0.49  | 1.70 $\pm$ 0.28  | 1.62 $\pm$ 0.52  | 1.54 $\pm$ 0.10   |
>
> **Q3:** The enhancement of CAT is limited.
>
> **A3**: Note that even a single successful attack can lead to substantial losses. When ASR increases from 1% to 10%, the incident frequency becomes 10 times larger, turning a negligible risk into a significant threat. In contrast, models with ACC of either 1% or 10% are both considered unusable in practice.
>
> In the following table, we have listed the average improvement of CAT against various defense algorithms under a given attack method, along with the changes in the incident frequency. It is evident that our method significantly increases the incident frequency. Notably, it even increases the incident frequency of BadNets to ~34 times larger compared to its original value.
>
> | ASR (Average over five defense)   | BadNets | Blend | CLB   | SIG   |
> | --------------------------------- | ------- | ----- | ----- | ----- |
> | Vanilla                           | 1.41    | 8.00  | 6.13  | 2.23  |
> | CAT                               | 47.80   | 46.28 | 14.18 | 27.18 |
> | Improvement of indecent frequency | 33.9x   | 5.8x  | 2.3x  | 12.2x |

---

> ### Author Response · Authors · 2023-11-20
> **Response to Reviewer FJ73 (Part2)**
>
> **Q4:** Some symbols used in the formulas lack explanations. Appendix C Figure 7 should refer to Table 7.
>
> **A4:**  Thank you for the correction. We have read the entire paper word by word and two minor errors have been corrected (The corrections are highlighted in blue.):
>
> - In Appendix C, we substitute “Figure 7” with “Table 7”. Figure 7 is correctly referred to and explained in Appendix G.
> - In Section 4, we correct “there exists” with “there exist”.
>
> We sincerely hope that our response could answer your questions and you could reconsider your rating. Look forward to your replies!
>
> [1] An Image is Worth 16$\times$16 Words: Transformer for Image Recognition at Scale, Dosovitskiy et al, in ICLR 2021.
>
> [2] BackdoorBench: A Comprehensive Benchmark of Backdoor Learning, Baoyuan Wu et al., In NeurIPS 2022.

---

> > ### Author Response · Authors · 2023-11-22
> > **Look forward to your reply**
> >
> > Dear Reviewer FJ73,
> >
> >  We would like to express our sincere gratitude to your insightful review. Hope our responses have successfully eliminate your doubts regarding the contribution of this paper, the reliability of the findings, the enhancement of CAT and some minor errors of this paper. As the final deadline of reviewer-author discussion is approaching, we still look forward to your valuable feedback if any concerns remain. We are ready to provide further elaboration and engage in a more in-depth discussion. If you are satisfied with our replies, please don't hesitate to update your score.
> >
> > Best wishes,
> >
> > Authors

---

### Meta-Review · Area_Chair_XhuY · 2023-12-10

**Metareview:**

The work focuses on VIT-based backdoor attacks. It highlights deficiencies in finetuning-based defense and pruning-based defense on ViT and emphasizes the importance of using AdamW for finetuning-based defense and only pruning the channel of the linear layer for pruning-based defense. Additionally, the authors introduce a new backdoor attack method, CAT, designed to bypass these defenses with increased robustness. CAT involves adding special adversarial perturbations to the trigger pattern to minimize noticeable channel activation differences between benign and triggered input.

Strengths:

All reviewers agree that this paper is well-written and easy to follow. The observation that optimizer plays an important role for ViT-based adversarial defense is interesting.

Weakness:

Most reviewers have highlighted the insufficient experiments in the initial reviews, such as the absence of certain baselines and the generalizability of CNNs. The authors responded by presenting additional experimental results during the rebuttal and effectively addressed some of the reviewers' concerns. However, reviewers still express concerns regarding the contribution of this work. Specifically, all the findings are derived from empirical studies without theoretical guarantees. Moreover, the issue of overfitting is a well-known concern in the adversarial community, and the proposed solutions lack substantial insights. We encourage the authors to consider the suggestions provided by the reviewers for incorporation into their work. As it stands, the current version does not meet the acceptance criteria of ICLR.

**Justification For Why Not Higher Score:**

The contribution of this work is limited without enough scientific insights to the community.

**Justification For Why Not Lower Score:**

N/A

---

### Decision · Program_Chairs · 2024-01-16

Reject